# Novel Insights into the Contribution of Cyclic Electron Flow to Cotton Bracts in Response to High Light

**DOI:** 10.3390/ijms24065589

**Published:** 2023-03-15

**Authors:** Xiafei Li, Weimin Ma, Wangfeng Zhang, Yali Zhang

**Affiliations:** 1The Key Laboratory of Oasis Eco-Agriculture, Xinjiang Production and Construction Group/College of Agronomy, Shihezi University, Shihezi 832003, China; 2College of Life Sciences, Shanghai Normal University, 100 Guilin Road, Shanghai 200234, China

**Keywords:** cotton bracts, photoprotection, cyclic electron flow, proton gradient across thylakoid membrane, heat dissipation

## Abstract

Cyclic electron flow around photosystem I (CEF-PSI) is shown to be an important protective mechanism to photosynthesis in cotton leaves. However, it is still unclear how CEF-PSI is regulated in non-foliar green photosynthetic tissues such as bracts. In order to learn more about the regulatory function of photoprotection in bracts, we investigated the CEF-PSI attributes in Yunnan 1 cotton genotypes (*Gossypium bar-badense* L.) between leaves and bracts. Our findings demonstrated that cotton bracts possessed PROTON GRADIENT REGULATION5 (PGR5)-mediated and the choroplastic NAD(P)H dehydrogenase (NDH)-mediated CEF-PSI by the same mechanism as leaves, albeit at a lower rate than in leaves. The ATP synthase activity of bracts was also lower, while the proton gradient across thylakoid membrane (ΔpH), rate of synthesis of zeaxanthin, and heat dissipation were higher than those of the leaves. These results imply that cotton leaves under high light conditions primarily depend on CEF to activate ATP synthase and optimize ATP/NADPH. In contrast, bracts mainly protect photosynthesis by establishing a ΔpH through CEF to stimulate the heat dissipation process.

## 1. Introduction

Photosynthates account for more than 90% of crops’ biomass [1]. Leaves are thought to be the main source of photosynthesis; in addition, numerous non-foliar tissues (i.e., rice panicle, glumes and lemmas of wheat, and cotton bracts) also exhibit stable photosynthetic characteristics and strong ability, especially in stressful environments [2,3,4,5,6,7,8]. The photosynthesis of non-foliar tissues in rice, which is quantitatively equivalent to a flag leaf, is correlated with the grain setting ratio [4]; non-foliar tissues of wheat have a stronger xanthophyll cycle which may contribute to higher resistance under adverse environments in later stages, enabling the plant to synthesize more photosynthate [2,3]. The photosynthesis of non-foliar tissues in cotton has a high contribution to the accumulation of assimilates at the late growth stage [5,6,7,8]. Thus, it is particularly important to explore the photosynthetic capacity and photoprotection mechanism of non-foliar tissues for high assimilates and yield.

Cotton bracts are main non-foliar tissues which produce a high contribution of photosynthates in the cotton plant [5,6,7,8]. Cyclic electron flow around photosystem I (CEF-PSI) is known to be an important protective mechanism to photosynthesis in cotton leaves [9]. The PSI-CEF works primarily through two pathways: PROTON GRADIENT REGULATION5 (PGR5) [10,11] and the choroplastic NAD(P)H dehydrogenase (NDH) complex [12,13,14,15]. Studies have shown that the NDH and PGR5 pathways play a critical part in the process of resisting adverse environmental conditions because they comprise two complementary CEF pathways that cannot both be absent at the same time [16,17]. The functional mechanism of cyclic electron flow in cotton leaves is clear, the PSI-CEF-driven proton gradient across the thylakoid membrane (ΔpH) can also regulate the ATP/NADPH ratio, which maintains the energy requirements of carbon absorption [18,19,20]. In addition, it can also promote non-photochemical quenching (NPQ), which reduces the amount of excessive energy absorbed and protects PSII from photoinhibition [9,19]. Regarding the protective mechanism in cotton bracts, previous studies have only shown that the strong ability of bracts to contribute to higher resistance under adverse environments is related to their heat dissipation mechanism [7], but there has been little research on the important protective mechanism of PSI-CEF that drives the stronger heat dissipation in bracts.

Cotton bracts have a lesser photosynthetic capacity than functional leaves, which indicates that cotton bracts will suffer more excess excitation energy at the same light intensity than leaves [21,22]. In addition, Hu et al. [21] has shown that the relative contribution of heat dissipation to yield of cotton bracts would rise under adverse environmental conditions. This further demonstrated the significance of cotton bracts’ photoprotective mechanisms. However, the mechanism by which non-photochemical quenching in cotton bracts stimulates the heat dissipation that aids in the steady operation of photosynthetic machinery is still unclear. Thus, it is necessary to explore the underlying mechanisms of photoprotection in cotton bracts by comparative analyses of the changes in CEF-PSI, ΔpH, and heat dissipation in cotton leaves and bracts under different light intensities.

## 2. Results

### 2.1. The Difference of Pigment Contents between Cotton Leaves and Bracts

Chlorophyll a (Chla) and chlorophyll b (Chlb) are crucial for the absorption, transmission, and conversion of light energy in photosynthesis in plants. However, carotenoids (Car) can not only capture light energy as auxiliary pigments but also quench active oxygen and prevent damage to the photosynthetic organs [23], and the level of Car/ChlT (the ratio of Car and total chlorophyll (ChlT)) is closely related to the ability of plants to tolerate adversity. Chla, Chlb, ChlT, and Car concentrations in cotton leaves were substantially higher (*p* < 0.05) than those in bracts. However, Chla/Chlb values in bracts were no different than those in leaves, indicating that both bracts and leaves have the ability to avoid the photoinhibition caused by excessive absorption of light energy (Table 1).

### 2.2. Cyclic Electron Flow Characteristics of Leaves and Bracts

Transcriptome sequencing showed that the expression levels of PGR5 and NDH in leaves were significantly higher (*p* < 0.05) than those in bracts; however, the expression level of PGRL1 showed no significant difference between leaves and bracts (Figure 1A). Furthermore, the expression of NdhM (NdhM is a subunit of NDH complex, and the amount of its protein accumulation can reflect the activity of NDH-mediated cyclic electron flow) and PGR5 proteins in leaves were higher (*p* < 0.05) than in bracts (Figure 1B). These indicated both NDH- and PGR5-mediated CEF pathways in leaves and bracts, and these occupied a large proportion in leaves compared with bracts.

We also measured CEF by monitoring the reduction rate of P700^+^ after illumination of leaves and bracts with far-red light. Induced by far-red light, P700 underwent charge separation and was oxidized. After the far-red light was turned off, the oxidized P700 was reduced by electrons in the electron transport chain. Thus, the speed of the decline slope of the P700 reduction curve indicated the rate of CEF, including NDH-mediated and PGR5-mediated PSI-CEF (Figure 2). Thus, the rate of CEF under high light was also limited, and the rate of CEF in leaves was significantly (*p* < 0.01) higher than in bracts (Figure 2B).

The transient Increase in chlorophyll fluorescence after the light was turned off, which was mainly caused by NDH-mediated CEF [24,25]. In low light, the NDH-mediated CEF in leaves was low or nearly nonexistent, whereas it was higher in bracts compared with leaves. Cotton leaves and bracts had identical NDH-mediated CEF when exposed to medium light, and the NDH -mediated CEF in leaves was still stimulated under high light (Figure 3).

### 2.3. Cotton Leaves Have a Higher ATP Synthase Activity Compared with Cotton Bracts

The P515 rapid relaxation kinetics of leaves and bracts were induced by a single-turnover saturating flash, which could reflect the integrity of the thylakoid membrane and the activity of ATP synthase [26]. After dark adaptation for 1 h of cotton leaves and bracts, the absorption signal of P515 increased rapidly at first, followed by a slow phase, and then decreased slowly after reaching the peak value. The signal curve of slow decline (Before AL) reflected the activity of the thylakoid membrane: the integrity of the leaves and bracts’ thylakoid membranes were high. Then the pre-illumination actinic light (2800 μmol m^−2^ s^−1^, 1800 μmol m^−2^ s^−1^, or 400 μmol m^−2^ s^−1^) was first turned on for 10 min, followed by dark adaption for 4 min (Figure 4). The slowly rising phase vanished as the P515 absorption signal grew quickly and then swiftly declined [27]. As shown in Figure 4, the initial slope of the rapidly declining P515 absorption signal after light pre-exposure can be used to determine ATPase activity. As the light intensity gradually increased from low to high, the initial slope of the rapidly declining phase of the leaves gradually increased, indicating that the ATP synthase activity of the leaves gradually increased, while the initial slope of the rapidly declining phase of bracts gradually decreased, indicating that the activity of ATP synthase in bracts gradually decreased with the increase in light intensity.

### 2.4. Heat Dissipation Characteristics of Leaves and Bracts

The relative zeaxanthin content and changes in ΔpH could be observed in the dark–light–dark, which caused a transient signal curve of the 550–515 nm signal [26]. The sample was totally dark adapted prior to the analysis to lower the zeaxanthin level. The formation of zeaxanthin can be reflected by an increase in light-induced signal. The formation of zeaxanthin is a slow process, it is formed gradually as protons accumulate in the thylakoid cavity. The relative extent of zeaxanthin formation can be judged from the increase in the “dark baseline” apparent after the actinic light is switched off. Due to the high stability of the 550–515 nm signal difference, the slow change in the dark baseline can be used to reliably evaluate the reversible zeaxanthin content change. The ΔpH can be determined [28] based on the change in curve after the actinic light was turned off. The signal above the black baseline gradually increased as the light intensity increased (Figure 5A), demonstrating that zeaxanthin synthesis was taking place. Higher zeaxanthin was generated in bracts than in leaves, as evidenced by the fact that the signal increase above the dark baseline of the bracts was much greater than that of the leaves. The magnitude of the steady-state ΔpH was determined by the difference between the dark baseline and the signal’s lowest point when the actinic light was turned off. With the increase in light intensity from low light to high light, the ΔpH gradually increased, especially in the medium and high light intensity; the ΔpH of the leaves was lower than that of the bracts. Zeaxanthin formation is catalyzed by the violaxanthin de-epoxidase that transforms violaxanthin through the intermediate antheraxanthin to zeaxanthin. When zeaxanthin is coupled with a light-harvesting protein, the protein’s structure changes, which causes the excitation energy to be quenched by heat dissipation (Table 1) [29]. According to further examination of the NPQ by the fluorescence quenching curve, under conditions of low and medium light, the NPQ of bracts was marginally greater than that of leaves. However, the NPQ of bracts was significantly higher than that of leaves at 2800 μmol m^−2^ s^−1^ high light (Figure 5B). The above results showed that the NPQ of bracts was significantly stronger under extremely high light conditions, accompanied by a strong ΔpH. These findings imply that NPQ is driven by the ΔpH of the bracts and that the xanthophyll cycle may be involved. It appeared that PSI-CEF in cotton leaves and bracts could promote the establishment of ΔpH (Figure 5), and then drive NPQ to dissipate excess light energy, especially in bracts.

## 3. Discussion

### 3.1. Bracts’ Photoprotection Mechanisms Differ from Those of Leaves, Which Primarily Rely on Establishing a Proton Gradient across the Thylakoid Membrane through Cyclic Electron Flow to Stimulate Heat Dissipation

The CEF-PSI plays an important role in plants’ resistance to adverse environmental conditions [10,16,17], which mainly includes CEF mediated by PGR5 [10,11] and NDH [12,13,14,15]. CEF-PSI can regulate the redox state of the photosynthetic electron transport chain, which promotes the establishment of a ΔpH across the thylakoid membrane, drives ATP synthase to synthesize ATP, and adjusts the ratio of ATP/NADPH [15,18] to stabilize carbon assimilation [10,17,19], while also promoting heat dissipation [30]. This study discovered that cotton leaves and bracts both have CEF-PSI routes that are mediated by PGR5 and NDH, but leaves have a higher level of CEF activity than bracts (Figure 1, Figure 2 and Figure 3). As a result, non-foliar green organs such as bracts, which are infrequently investigated by scientists, also have a complete NDH-mediated and PGR5-mediated PSI-CEF, which can play a role in regulating the photosynthetic electron transport chain similar to that in leaves.

The rate of PSI-CEF increased in cotton leaves as the light intensity steadily increased from low light to high light. The activity of ATP synthase also increased (Figure 2, Figure 3 and Figure 4), which was conducive to CO_2_ fixation. At the same time, the leaves had a high chlorophyll content (Table 1) and a strong light absorption capacity; thus, they would suffer more excess excitation energy, leading to the speed of photodamage exceeding the speed of repair. The excited CEF could drive ATP synthase to synthesize ATP for the repair of PSII photodamage (Figure 2, Figure 3 and Figure 4). Therefore, leaves mainly relied on cyclic electron flow to form ATP and to regulate the ATP/NADPH ratio.

The ΔpH could drive the NPQ [9,10], and bracts had stronger NPQ capabilities than leaves according to both earlier research and the current study [21,22]. Correspondingly, under medium and high light conditions, the ΔpH and zeaxanthin synthesis rates of bracts were significantly higher than those of leaves (Figure 5). Therefore, bracts mainly relied on the establishment of ΔpH to stimulate NPQ. The production of ΔpH was also regulated by CEF-PSI, and the higher ΔpH of bracts was mainly related to complete NDH-mediated and PGR5-mediated PSI-CEF (Figure 1, Figure 2, Figure 3 and Figure 5A). The ability of cotton bracts to perform photosynthetic phosphorylation decreased as light intensity increased from low light to high light. Additionally, research has shown that photorespiration could be weak in bracts living in high carbon dioxide environments, which results in the corresponding oxidation state [21]. As a result, excess excitation energy requires more NPQ to dissipate excitation energy and protect PSII from photoinhibition. Studies have shown that alternative electron transport pathways, ion transporters/channels, pH-dependence [31] and plastid terminal oxidase could also regulate the production of ΔpH [15]; thus, whether the bracts have other electron transport pathways that are involved in the regulation of NPQ is worthy of further study. However, the generation of heat dissipation indicates that the photochemical activity of the photosynthetic mechanism is weak and generates more excess excitation energy. On the other hand, it also indicates that the photosynthetic apparatus itself has the ability to dissipate the excess excitation energy safely and harmlessly [32]. Bracts obviously have a lower photosynthetic potential than leaves (Table 1). Due to the low photosynthetic capacity of bracts, they experience a greater excitation energy at the same level of light intensity, and thus bracts stimulate a stronger NPQ capacity to maintain the stable operation of the photosynthetic apparatus. The results illustrated in Figure 3, Figure 4 and Figure 6 show that ΔpH, heat dissipation, and CEF increased in response to low light to high light, respectively. This suggests that the bracts primarily depend upon the ΔpH established by CEF to drive the NPQ to dissipate excess light energy. However, under medium to high light, the ΔpH reached a stable status and the heat dissipation capacity remained unchanged. It can be speculated that under high light conditions, although the excess excitation energy of the leaves increased, the ability of the leaves to protect the photosynthetic mechanism through heat dissipation in high light was weakened. However, under high light, the heat dissipation process of the bracts can be stably operated to protect the photosynthetic machinery. This result further confirms our earlier observation that bracts’ photosynthetic systems could withstand significant stress in challenging environments [7].

### 3.2. The Importance of Heat Dissipation Capacity in Bracts in Response to High Light

To respond to environmental changes, different species have evolved different photoprotection mechanisms [17,33]. This suggests that there are also differences in photoprotective mechanisms between cotton leaves and non-foliar green organs such as cotton bracts. The primary sites for photosynthesis are cotton leaves because of their abundant chlorophyll content and powerful photophosphorylation ability. During the long-term evolution process, leaves mainly relied on CEF to drive ATP synthase to synthesize ATP and optimize ATP/NADPH for efficient photosynthesis (Figure 4). Due to the minimal amount of light energy that was absorbed, transmitted, and converted by the cotton bracts, they had a low level of photosynthetic phosphorylation ability. The ΔpH could be established to stimulate heat dissipation which maintained the redox state of the photosynthetic electron transport chain (Figure 6B). While this did not occur in the cotton leaves, cotton bracts could constantly increase their capacity to dissipate excess light energy under high light (Figure 5). These data indicated that cotton bracts may be of great significance in compensating for the reduction in leaf carbon assimilation capacity, thereby maintaining carbon assimilation and life activity in cotton plants. In addition, with the continuation or aggravation of the stress environment (high light) in the flowering and boll stages of cotton [7], the light captured by cotton leaves was not entirely used for carbon assimilation. Excess light energy led to the production of reactive oxygen species, which harmed the photosynthetic organs and affected the stable operation of leaves’ photosynthesis, even affecting cotton yield. Meanwhile, the heat dissipation ability of the cotton bracts was strong, which prevented the accumulation of reactive oxygen species and maintained the carbon assimilation ability of cotton plants, thus contributing to the increase in yield. According to earlier studies [5,6,7,8] and our results, cotton bracts exhibit a completely functional photosynthetic apparatus, as well as a strong capacity to withstand environmental stress (Figure 5). As a result, cotton bracts could serve as a supplemental carbon source for the photosynthetic capacity of cotton plants, which is of great significance in increasing the photosynthetic capacity of cotton [6]. Therefore, when selecting and breeding cotton varieties in the future, it is important to focus on the excellent non-foliar green organ traits. Non-foliar green organ bracts can be used as a stress-resistant indicator of variety, and they can also be fully utilized for their high photosynthetic efficiency in environmental conditions and achieving a high and stable cotton yield.

In conclusions, non-foliar green organs bracts exhibit a relatively stable photosynthetic capacity in high environmental stress conditions. To maximize the total photosynthetic capacity of cotton, we need to take into account the contributions of PSI-CEF in cotton bracts.

## 4. Materials and Methods

### 4.1. Plant Material

The experiments were conducted at the experimental farm of Shihezi University (44°17′ N, 86°03′ E), Xinjiang, China, in 2018. The cultivar of Yunnan 1 (*Gossypium barbadense* L.) was assigned to three plots and planted on 21 April 2018; drip irrigation tubes were installed on 22 April 2018 followed by manual sowing after film coating, and 120 m^3^ ha^−1^ of irrigation was carried out two days later to ensure the emergence of seedlings. A wide–narrow row-planting pattern (66 + 10 cm) with a plant spacing of 12 cm was used. The size of each plot area was 4.5 m^2^, and each plot was uniformly fertilized with 150 kg ha^−1^ of compound fertilizer (15% N, 15% P, 15% K) and 240 kg ha^−1^ (urea, 46% N) before sowing, and an additional 260 kg ha^−1^ of compound fertilizer was applied during the flowering stage and boll stage. Weeds and pests were controlled using local standard management practice.

The experimental materials were sampled at the maturity stage, and the highest light intensity during the stage was 1500–2000 mol m^−2^ s^−1^. The first fully expanded leaves and the bracts corresponding to the first fruit node of each cotton plant were sampled, and three cotton plants were sampled in each plot. The sampling method is shown in Figure 7.

### 4.2. Chlorophyll Content

The cotton leaves and bracts were punched into 4 small discs with a perforator (diameter 5 mm) before placing them into a 5 mL centrifuge tube. Chlorophyll was extracted using 4 mL 80% acetone solution in low light. A UV-2041 spectrophotometer (Shimadzu Co., Kyoto, Japan) was used to colorimetrically measure the supernatant. According to the method of Lichtenthaler [34], the contents of Chla, Chlb, ChlT, and Car were determined.

### 4.3. Transcriptome Sequencing

The RNA extraction, quality detection, library construction, and sequencing of cotton leaves and bracts were completed by Beijing Novogene Bioinformation Technology Co. (Beijing, China).

An Illumina HiSep 4000 was used to double-terminally sequence the RNA samples (Beijing Novogene Bioinformation Technology Co., Beijing, China). The height of transcripts in the sample is indicated by the FPKM (fragments per kilobase of transcript per million fragments mapped) number. The reference standard expression differences (FC, the fold change) were set to be greater than 4, or |FC| > 4, and *p* < 0.05 was the significantly differentially expressed gene.

### 4.4. Isolation of Chloroplasts

Samples of leaves and bracts were homogenized in a medium containing 330 mM sorbitol, 20 mM Tricine/NaOH (pH 7.6), 5 mM EGTA, 5 mM EDTA, 10 mM NaHCO_3_, 0.1% (*w*/*v*) BSA, and 5 mM ascorbate. After centrifugation for 5 min at 2000× *g*, the pellet was resuspended in 300 mM sorbitol, 20 mM Hepes/KOH (pH 7.6), 5 mM MgCl_2_, and 2.5 mM EDTA. Intact chloroplasts were purified by passing through 40% Percoll.

### 4.5. Sodium Dodecyl Sulfate Polyacrylamide Gel Electrophoresis (SDS-PAGE)

Chloroplasts were osmotically ruptured in buffer containing 20 mM HEPES/KOH (pH 7.6), 5 mM MgCl_2_, and 2.5 mM EDTA. Thylakoid membranes were separated from the stromal fraction by centrifugation (15,000× *g* for 10 min at 4 °C). The chlorophyll content was determined by the method of Porra et al. [35].

SDS-PAGE of thylakoid membranes was carried out on 12% polyacrylamide gel with 6 M urea as described earlier [36]. The coarse thylakoids extracted in accordance with 4.4 were resuspended with medium buffer solution, adjusted to the same chlorophyll concentration, and then divided into 1.5 mL centrifugal tubes according to the required loading amount. A quarter volume of 5 × SDS-loading buffer (2 M urea, 0.5 M Tris-HCl, respectively) was added. A solution of pH 8.0, 20% glycerin, 7.5% SDS, 2% (*v*/*v*) mercaptoethanol, 0.05% (*w*/*v*) bromophenol blue was added, and samples were boiled in a water bath at 100 °C for 8 min, and then centrifuged at room temperature at 12,000× *g* for 10 min. The supernatant was transferred into a new centrifugal tube and samples were taken. Samples were allowed to run out of the wells at a voltage of 60 V, and the voltage was adjusted to 80 V until the end of electrophoresis.

### 4.6. Western Blotting Analysis of Proteins

Protein Western blotting was performed after electrophoresis. The proteins in the polyacrylamide gel were electrotransferred to a polyvinylidene difluoride membrane (Immobilon-P Millipore Co., Bedford, UK), and the PVDF membrane was placed in 5% skim milk prepared by TBST and closed in a shaker for 2 h at room temperature. The corresponding protein-specific antibodies were incubated in the shaker for 2 h, and then TBST was used to wash the antibodies twice for 12 min per wash. Then, the corresponding dilution of the second antibody was added and incubated for 1 h. TBST was used to wash the antibody four times for 12 min per wash. The development test was performed according to the ECL assay kit (Amersham Pharmacia Biotech., Piscataway, NJ, USA). Antibodies against NAD(P)H-quinone oxidoreductase subunit M (NdhM) (1:1000) and PGR5-like photosynthetic phenotype 1 (PGRL1) (1:1000) were provided by Professor Lianwei Peng, College of Life Sciences, Shanghai Normal University, Shanghai, China. This experiment was carried out with the help of Professor Weimin Ma, College of Life Sciences, Shanghai Normal University, Shanghai, China.

### 4.7. P700 Dark Reduction Curve

Leaves and bracts were exposed to actinic light (620 nm, 400 μmol m^−2^ s^−1^, 1800 μmol m^−2^ s^−1^, or 2800 μmol m^−2^ s^−1^), then exposed to far-red light (>705 nm, 5.2 μmol m^−2^ s^−1^) for 40 s using a Dual-PAM-100 chlorophyll modulation fluorescence instrument (Walz Co., Effeltrich, Germany). The P700^+^ was re-reduced after the far-red light was turned off. According to the change in redox kinetics, the oxidation level of P700 and the initial slope of reduction were analyzed to obtain the initial rate of 810–830 nm light absorption decrease after the far-red light was turned off.

The formula for calculating the initial slope was (V_0_ − V_0.2_)/0.2. V_0_ referred to the voltage value at the instant when the far-red light was turned off, and V_0.2_ referred to the voltage value at 0.2 s after turning off the far-red light [12].

### 4.8. Post-Illumination Chlorophyll Fluorescence Transient

After dark adaptation for 30 min, post-illumination chlorophyll fluorescence transients were measured from leaves and bracts using the Dual-PAM-100 chlorophyll modulation fluorescence instrument (Walz Co., Effeltrich, Germany). They were first illuminated for 4 min at (400 μmol m^−2^ s^−1^, 1800 μmol m^−2^ s^−1^, or 2800 μmol m^−2^ s^−1^). After the fluorescence kinetics were induced, the actinic light was turned off. The prompt increase in chlorophyll fluorescence after turning off the actinic light was used to reflect redox changes of the PQ pool [24].

### 4.9. P515 Signal Changes

The new module P515/535 of a Dual-PAM-100 (Walz Co., Effeltrich, Germany) was used to detect the differential absorption of 550–515 nm measuring light. Firstly, the integrity of the thylakoid membrane was measured after dark adaptation for 60 min, and then the activity of ATP synthesis was determined after being exposed to actinic light (620 nm, 400 μmol m^−2^ s^−1^, 1800 μmol m^−2^ s^−1^, or 2800 μmol m^−2^ s^−1^) for 10 min, then samples were dark adapted for 4 min [26].

The dark–light–dark-induced transients of 550–515 nm signal can reflect proton gradient across the thylakoid membrane. Before the change in the P515 signal was measured, the experimental material was dark adapted for a few hours, and then the actinic light (400 μmol m^−2^ s^−1^, 1800 μmol m^−2^ s^−1^, or 2800 μmolm^−2^ s^−1^) was turned on and induced for 10 min, and the actinic light was turned off and stopped when the P515 signal stabilized.

### 4.10. Non-Photochemical Quenching (NPQ)

A Dual-PAM-100 chlorophyll fluorescence instrument (Walz Co., Effeltrich, Germany) was used to determine the chlorophyll fluorescence parameters. After the leaves and bracts were dark adapted for 30 min, the initial fluorescent (Fo) and the maximum fluorescent (Fm) were measured, then the actinic light (400 μmol m^−2^ s^−1^, 1800 μmol m^−2^ s^−1^, or 2800 μmol m^−2^ s^−1^) was turned on, and a saturating light pulse light (20,000 μmol m^−1^ s^−1^, the duration of saturated pulse Fm detection was 500 ms) was turned on every 30 s. After 6 min, the actinic light was turned off for 4 min to observe the recovery of NPQ in the dark. NPQ was calculated according to the formula (Fm − Fm’)/Fm’ [37].

### 4.11. Statistical Analysis

Data processing was performed using Microsoft Excel 2019. Figures were plotted with R version 4.2.1 (https://www.r-project.org/, accessed on 1 March 2022). Least-significant difference (LSD) analyses were performed in the “agricolae” package of R version 4.2.1, and the differences between leaves and bracts were defined at *p* < 0.05 and *p* < 0.01. The data in tables and figures represent the mean (*n* = 3) ± SD.

## 5. Conclusions

Cotton leaves and bracts exhibit different photoprotective mechanisms in response to high light. Leaves mainly rely on cyclic electron flow to drive ATP synthase to synthesize ATP and optimize ATP/NADPH for efficient photosynthesis. In contrast, bracts can protect photosynthesis by establishing a ΔpH through cyclic electron flow to stimulate the process of heat dissipation, especially in high light; the heat dissipation ability of bracts can be continuously enhanced, while this was not observed in the leaves. Therefore, cotton bracts have a comparatively steady photosynthetic capability and stronger stress tolerance compared with leaves.

## Figures and Tables

**Figure 1 ijms-24-05589-f001:**
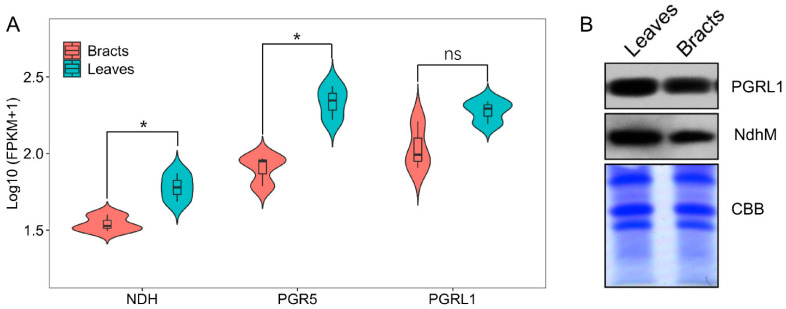
Differences in transcription and protein levels of cyclic electron flow. (**A**) Differences in the expression of electron-related genes in cotton leaves and bracts. Solid lines within the box indicate the mean values, box boundaries indicate upper and lower quartiles; The width of spindles denotes the accumulation of high samples. Bars indicate SD (*n* = 3). “*” and “ns” denotes significance at the 0.05 level and non-significant between the leaves and bracts based on an analysis of variance, respectively. (**B**) Coomassie brilliant blue (CBB)-staining PAGE profiles and Western analysis of total PGR5-like photosynthetic phenotype 1 (PGRL1) (1:1000) and NAD(P)H-quinone oxidoreductase subunit M (NdhM) (1:1000) in the thylakoid membranes of cotton leaves and bracts. Lanes were loaded with thylakoid membrane proteins corresponding to 1 μg and 10 μg chlorophyll for Western analysis and CBB, respectively.

**Figure 2 ijms-24-05589-f002:**
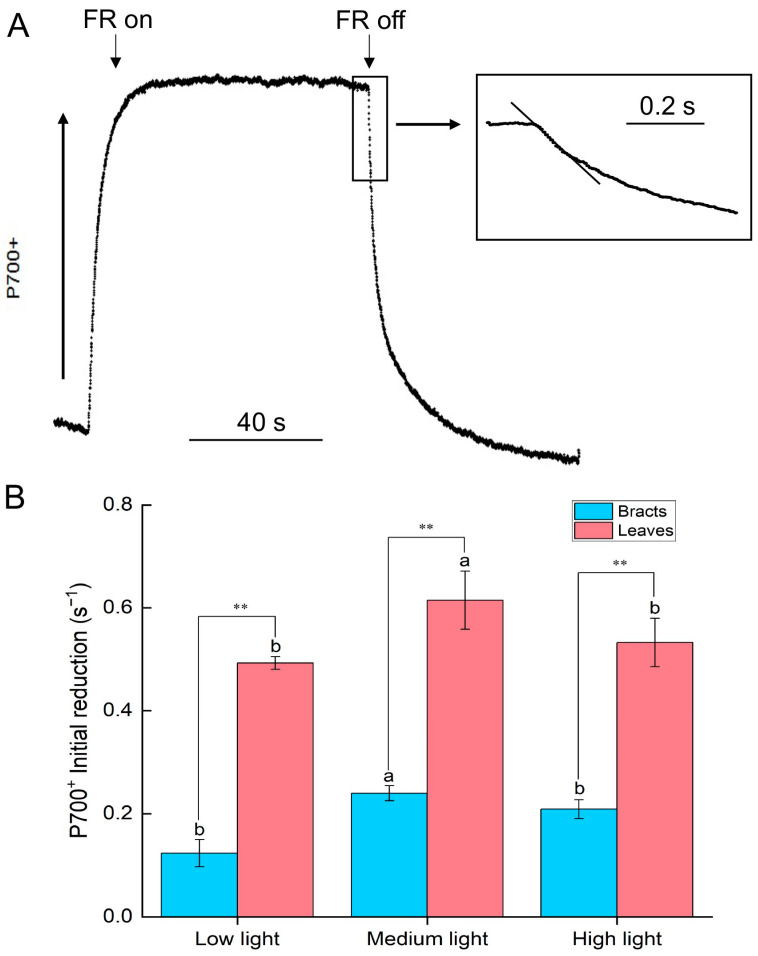
Effect of different light intensities on the initial reduction rate (0–0.2 s) of P700^+^ after turning off the far-red light (FR) in leaves and bracts, black rectangle reflects the initial reduction rate (0–0.2 s) of P700^+^ after turning off the far-red light as follows. (**A**) Dark reduction kinetics of P700^+^ with far-red light off (λ > 705 nm, 5.2 μmol m^−2^ s^−1^, lasted for 40 s); (**B**) P700^+^ initial reduction rate (0–0.2 s) under different PPFDs (400 μmol m^−2^ s^−1^, low light; 1800 μmol m^−2^ s^−1^, medium light; 2800 μmol m^−2^ s^−1^, high light). Different letters indicate a significant difference (*p* < 0.05) between the low, medium, and high light. ** denotes significance at the 0.01 level between the leaves and bracts based on an analysis of variance. Bars indicate SD (*n* = 3).

**Figure 3 ijms-24-05589-f003:**
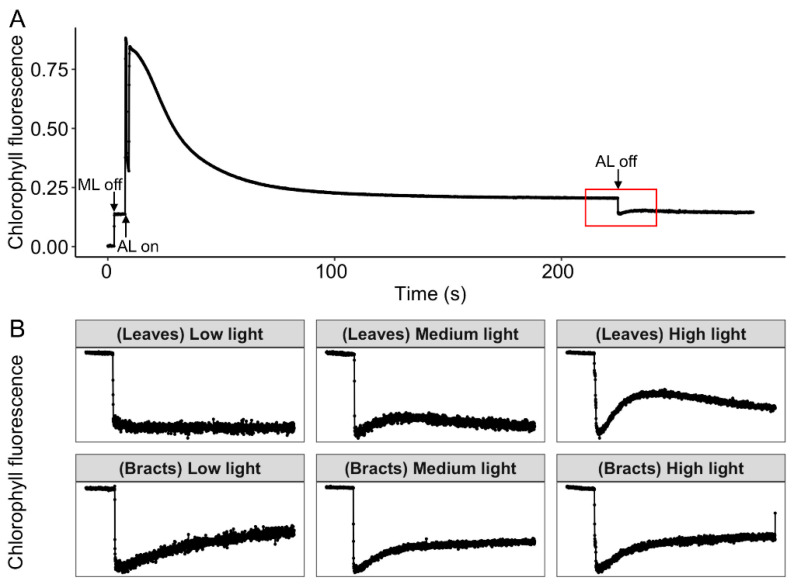
Monitoring of NDH activity (red rectangle below (**B**)) using chlorophyll fluorescence analysis in cotton leaves and bracts (*n* = 3) under different light intensities of 400 μmol m^−2^ s^−1^ (low light), 1800 μmol m^−2^ s^−1^ (medium light), or 2800 μmol m^−2^ s^−1^ (high light). (**A**) Induction kinetics and post-illumination transient in chlorophyll fluorescence: AL (actinic light) (620 nm, 400 μmol m^−2^ s^−1^, 1800 μmol m^−2^ s^−1^, or 2800 μmol m^−2^ s^−1^, lasted for 4 min), ML (measuring light). (**B**) Analyses of post-illumination chlorophyll fluorescence transient in cotton leaves and bracts.

**Figure 4 ijms-24-05589-f004:**
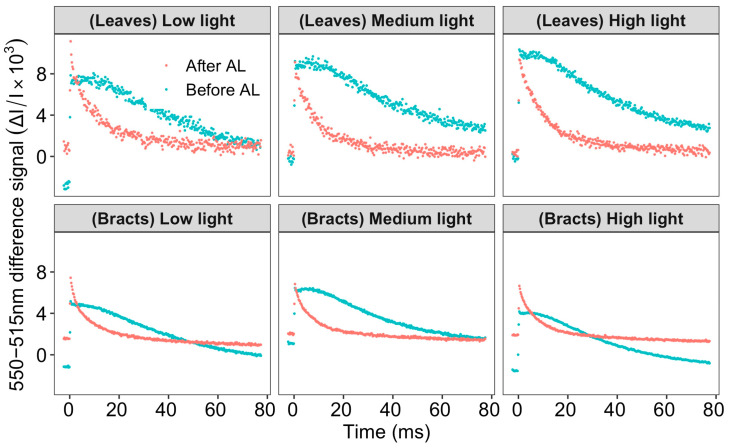
Changes in the P515 signal when a single-turnover saturating flash was applied to cotton leaves and bracts (*n* = 3) that had been given pre-illumination at different light intensities of 400 μmol m^−2^ s^−1^ (low light), 1800 μmol m^−2^ s^−1^ (medium light), or 2800 μmol m^−2^ s^−1^ (high light). After AL: the actinic light was turned on for 10 min, and then turned off, followed by dark adaptation for 4 min before measurement. Before AL: measurement of leaves and bracts after 1 h dark adaptation.

**Figure 5 ijms-24-05589-f005:**
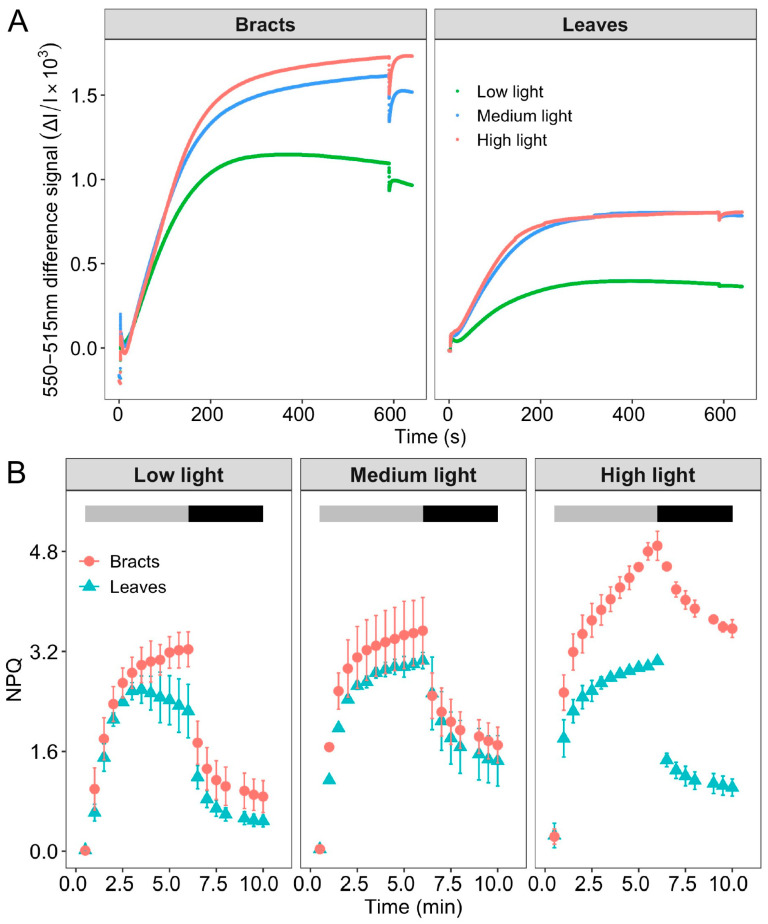
Changes in heat dissipation in cotton leaves and bracts after exposure to different durations of three different light intensities of 400 μmol m^−2^ s^−1^ (low light), 1800 μmol m^−2^ s^−1^ (medium light), or 2800 μmol m^−2^ s^−1^ (high light). (**A**) Slow dark–light–dark induction transients of the 550–515 nm signal in leaves and bracts in different light intensities. (**B**) Changes of NPQ in leaves and bracts under different light intensities. Bars indicate mean and SD (*n* = 3).

**Figure 6 ijms-24-05589-f006:**
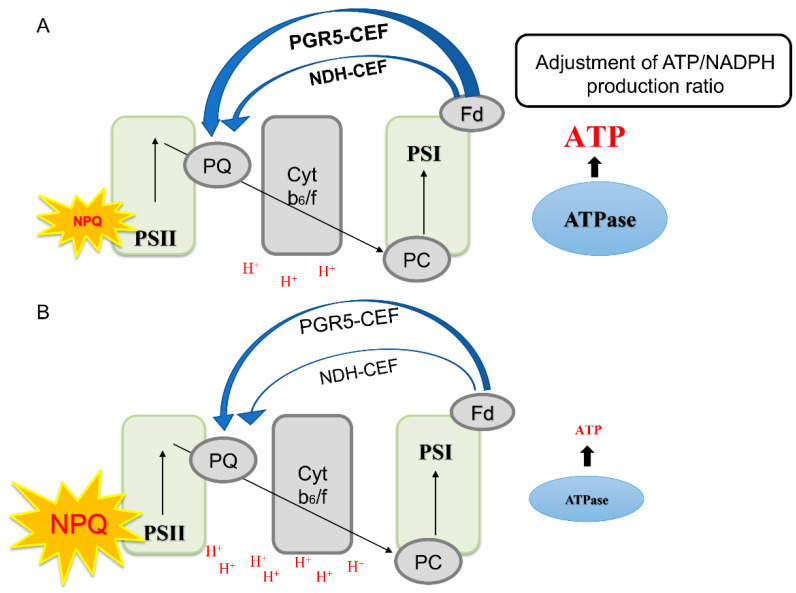
Different photoprotection mechanisms were induced in leaves (**A**) and bracts (**B**) from the same cotton plant. (**A**) The cotton leaves mainly depend on cyclic electron to optimize ATP/NADPH. (**B**) The cotton bracts mainly rely on the dissipation of excess excitation energy as heat driven by a proton gradient across the thylakoid membrane. The thickness of the arrow indicates the strength of the cyclic electron flow; ATP and NPQ font size and boldness reflect the strength of these functions.

**Figure 7 ijms-24-05589-f007:**
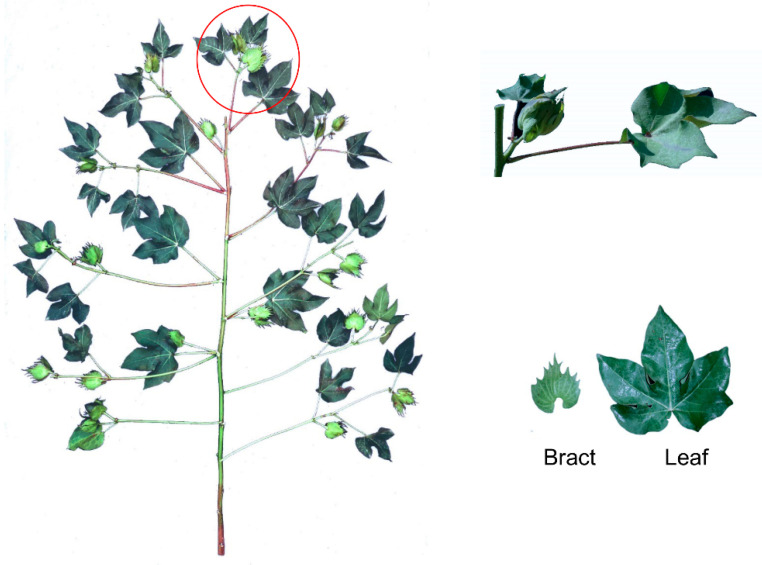
Schematic diagram of cotton leaves and bracts with sampling shown in red circle.

**Table 1 ijms-24-05589-t001:** The difference of photosynthetic parameter between cotton leaves and bracts.

Treatments	Chla (μmol dm^−2^)	Chlb (μmol dm^−2^)	Chla/Chlb	Car (μmol dm^−2^)	ChlT (μmol dm^−2^)	Car/ChlT
Leaves	6.64 ± 0.66 a	2.33 ± 0.3 a	2.85 ± 0.07 a	1.57 ± 0.04 a	8.98 ± 0.96 a	0.17 ± 0.02 b
Bracts	0.89 ± 0.08 b	0.31 ± 0.04 b	2.92 ± 0.15 a	0.56 ± 0.56 b	1.19 ± 0.12 b	0.47 ± 0.03 a

Different letters in the table indicate significant differences between leaves and bracts at the 0.05 level. Chla, Chlb, ChlT, and Car represent chlorophyll a, chlorophyll b, total chlorophyll, and carotenoids. Values in the table indicate the mean ± SD (*n* = 3).

## Data Availability

Data sharing not applicable.

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
