# Peer review of "Novel Insights into the Contribution of Cyclic Electron Flow to Cotton Bracts in Response to High Light"

_ijms, 2023, doi:10.3390/ijms24065589_

Round 1
Reviewer 1 Report (New Reviewer)
Article review: Xiafei Li, Weimin Ma, Wangfeng Zhang and Yali Zhang "Novel insights into the contribution of cyclic electron flow to cotton bract in response to high light"
The authors' article concerns the study of the role of cyclic electron flux of chloroplasts (CEF-PSI) of leaves and bracts in cotton genotypes in protecting photosynthesis to high light intensity conditions.
The results obtained showed that cotton leaves in high light conditions primarily depend on cyclic electron flow in the activation of ATP synthase and optimization of ATP/NADPH. Bracts mainly protect photosynthesis by setting ΔpH through
cyclic electron flow to stimulate the heat dissipation process.
The results obtained by the authors are new both in terms of assessing the role of CEF-PSI in cotton bracts, and clarifying the regulatory function of photoprotection mediated by chor-14, plastic NAD(P)H-dehydrogenase (NDH) -CEF-PSI by the same mechanism as the leaves.
I believe that the article can be accepted for publication
Author Response
We sincerely thank the reviewers for their time and consideration.
Reviewer 2 Report (New Reviewer)
The manuscript by Li et al. is devoted to the study of cyclic electron transfer around photosystem (PS) I under different light intensity in leaves and bract of cotton plants. The authors used many different approaches to investigate the cyclic electron transfer and made conclusions based on the obtained results. However, I have doubts that some results have been interpreted correctly and, therefore, the conclusions can be wrong. Thus, I think that the manuscript in the current view can not be published and is required significant revision.
I list my main comment below.
First of all, the Introduction is quite short and does not describe known data about the cyclic electron transfer obtained in non-foliar organs in other plants, including corticular photosynthesis. The novelty should be written more clearly.
In table 1 there is no Chla/Chlb ratio, but this parameter clearly indicates the changes in antenna size.
I disagree that the expression levels of PGRL1 in leaves were slightly higher (L.83) because there is no statistically significant difference in Fig.2A (the authors indicate ns).
What is NdhM? (L.84)
The concentration of which Chl the authors used in Blots, Chl a (indicated in the description) or total Chl, which is usually used? The picture of CBB is not related to 1 μg of Chl according to my experience in that. Can it be that the authors used 10 μg of Chl? Concerning to experiments below, for example with NPQ, why did the authors not make Blots for D1 (PsbA) protein of PSII? In addition, the repetition of blots with statistical processing are required, of course.
I disagree that the rate of CEF under medium light and high light were higher than that under low light (L101). The Fig 3B indicates the higher rate at medium light and the same rates at low and higher light. Probably, this would be correct to indicate a, b for leaves and a’, b’ for bracts as statistical differences.
I can not find the wavelength of the measured and actinic light used in Fig. 4.
The black baseline (L156) is not indicated in Fig. 6A.
I think that the authors made the incorrect measurements (as well as conclusions) of ΔpH because the normalization on flesh induced PSII+PSI signal is required (see New accessory for the DUAL-PAM-100: The P515/535 module and examples of its application, Ulrich Schreiber and Christof Klughammer, 2008) due to possible different content of PSII and/or I in leaves and bracts. In addition, how the authors can explain the lower ΔpH in leaves (L.163 and Fig 6A) but the significantly higher rate of CEF in leaves (L.102 and Fig. 3B), which should increase ΔpH.
I also doubt about NPQ results. The authors indicate that bracts had a considerably greater NPQ than leaves when light intensity increased, especially when there was a high light induction (L.168,169), however there are no statistically significant differences between NPQ induced in leaves and bracts at low and medium light (Fig. 6B left and center panels), while there is clear photoinhibition of PSII in bracts at high light (Fig. 6B right panel). Latter, of course, do not imply that the xanthophyll cycle's NPQ is driven by the ΔpH of the bracts (L170) as the authors write. See also in L205-206; L222-223; L232; L249.
This is well known that the content of PTOX in thylakoids is very small, about 1% compared to PSII components (see DOI 10.1007/s00425-003-1111-7), thus the involvement of PTOX in described effects can not to be so sufficient (L215).
The authors should indicate the duration of saturation pulse for Fm detection (200–500 ms) (L385)
Minor comments
L.259. () the author should insert the citations here.
L.357-358. Does Voltage mean value?
L.374–376. Confused sentences.
Author Response
Dear Editor,
We sincerely thank the reviewers for their comments regarding our manuscript. We have carefully revised the manuscript according to their suggestions. In our point-by-point response below, the reviewers’ comments are in Black and our responses are in BLUE. The modifications in the revised manuscript are in RED. We hope these modifications are acceptable to you. Thank you again for your time and consideration.
Reviewers/Editor comments:
- The Introduction is quite short and does not describe known data about the cyclic electron transfer obtained in non-foliar organs in other plants, including corticular photosynthesis. The novelty should be written more clearly.
Response: Thank you for your comments. We have modified introduction on 25-64 Lines. As follows:
Photosynthates account for more than 90% of crops biomass [1]. Leaves are thought to be the main source of photosynthesis; besides, numerous non-foliar tissues (i.e. rice panicle, glumes and lemmas of wheat and cotton bracts) also exhibit stable photosynthetic characteristics and strong ability, especially in stress environment [2-8]. Photosynthesis of non-foliar tissues in rice, which is quantitatively equivalent to a flag leaf is correlated with grain setting ratio [4]; non-foliar tissues of wheat have stronger xanthophyll cycle may contribute to the higher resistance under adverse environments in later stages to synthesize more photosynthate [2-3]. Photosynthesis of non-foliar tissues in cotton have high contribution to accumulation of assimilates at the late growth stage [5–8]. Thus, it is particularly important to explore the photosynthetic capacity and photoprotection mechanism of non-foliar tissues for high assimilates and yield.
Cotton bracts are main non-foliar tissues and have high contribution of photo-synthates to cotton plant [5-8]. Cyclic electron flow around photosystem I (CEF-PSI) is shown to be an important protective mechanism to photosynthesis in cotton leaves [9]. The PSI-CEF works primarily through PROTON GRADIENT REGULATION5 (PGR5) [10-11] and choroplastic NAD(P)H dehydrogenase (NDH) complex [12-15] two pathways. Studies have shown that the NDH and PGR5 pathways play a critical part in the process of resisting adverse environmental conditions because they comprise of two complementary CEF pathways that can not both be absent at the same time [16-17]. The functional mechanism of cyclic electron flow in cotton leaves is clear, PSI-CEF-driven a proton gradient across the thylakoid membrane (ΔpH) can also reg-ulate the ATP/NADPH ratio, which maintain the energy requirement of carbon absorption [18-20]. In addition, it can also promote non-photochemical quenching (NPQ), which reduce the amount of excessive energy absorbed and protect PSII from photoinhibition [9,19]. About the protective mechanism in cotton bracts, previous studies have only shown that the strong ability of bracts to the higher resistance under adverse en-vironments is related to their heat dissipation mechanism [7], but there has been little research on the important protective mechanism of PSI-CEF that drives the stronger heat dissipation in bracts.
The cotton bracts have a lesser photosynthetic capacity than the functional leaves, which indicate that cotton bracts will suffer more excess excitation energy at the same light intensity than leaves [21-22]. In addition, Hu et al. [21] has shown that the relative contribution of heat dissipation to yield of cotton bracts would rise under adverse en-vironmental conditions. This further demonstrated the significance of cotton bracts' photoprotective mechanisms. However, the mechanism of non-photochemical quenching in cotton bracts stimulate heat dissipation that aids in the steady operation of photosynthetic machinery was still unclear. Thus, it necessary to explore the under-lying mechanisms of photoprotection in cotton bracts by comparative analyzing the changes of the CEF-PSI, ΔpH and heat dissipation in cotton leaves and bracts under different light intensity.
- In table 1 there is no Chla/Chlb ratio, but this parameter clearly indicates the changes in antenna size.
Response: Thank you for your comments. We have modified that according to reviewer’s comment in lines 72-74. As follows: However, Chla/Chlb in bracts were not difference than that in leaves, this indicated that both bracts and leaves had the ability to avoid photoinhibition caused by excessive absorption of light energy.
- I disagree that the expression levels of PGRL1 in leaves were slightly higher (L.83) because there is no statistically significant difference in Fig.2A (the authors indicate ns).
Response: Thank you for your comments. We have modified this part of the content on 82-83 Lines. As follows: however, the expression level of PGRL1 had no significant difference between leaves and bracts (Figure 2A).
- What is NdhM? (L.84)
Response: Thank you for your comments. NdhM is a subunit of NDH complex, and the amount of its protein accumulation in Figure 2 can reflect the activity of NDH-mediated cyclic electron flow.
- The concentration of which Chl the authors used in Blots, Chla (indicated in the description) or total Chl, which is usually used? The picture of CBB is not related to 1 μg of Chl according to my experience in that. Can it be that the authors used 10 μg of Chl? Concerning to experiments below, for example with NPQ, why did the authors not make Blots for D1 (PsbA) protein of PSII? In addition, the repetition of blots with statistical processing are required, of course.
Response: Thank you for your comments. This part of the experiment from Shanghai Normal University Professor Weimin Ma's laboratory, therefore, we further confirmed the experimental results on 94-95 Lines. As follows: (CBB) Lanes were loaded with thylakoid membrane proteins corresponding to 10 μg chlorophyll a.
Western blot on 348-349 Lines, as follows: Lanes were loaded with thylakoid membrane proteins corresponding to 1 μg chlorophyll a.
We are very sorry that our laboratory does not have the conditions for biochemical experiments. Secondly, our coauthor Weimin Ma's research group does not have the antibodies related to NPQ.
- I disagree that the rate of CEF under medium light and high light were higher than that under low light (L101).
Response:Thank you for your comments. We have modified that according to reviewer’s comment in lines 100-102. As follows: Thus, the rate of CEF under high light was also limited, and the rate of CEF in leaves were sig-nificantly (P < 0.01) higher than in bracts (Figure 3B).
- The Fig 3B indicates the higher rate at medium light and the same rates at low and higher light. Probably, this would be correct to indicate a, b for leaves and a’, b’ for bracts as statistical differences.
Response: Thank you very much for your careful examination and comments; the bracts and leaves are only included in the same light, and the value in leaves were higher than bracts, thus we chose the analysis of variance to analyze the difference. We have also added the notes in figure title, as follows: ** denotes significant at the 0.01 level between the leaves and bracts based on analysis of variance. If we misunderstand the meaning of the reviewer, we can revise it again.
- I can not find the wavelength of the measured and actinic light used in Fig. 4.
Response: Thank you for your comments. We have modified this part of the content on 123 Line, 373 Line. As follows: AL (actinic light) (620 nm, 400 μmol m-2 s-1, 1800 μmol m-2 s-1 or 2800 μmol m-2 s-1, lasted for 4min).
- The black baseline (L156) is not indicated in Fig. 6A.
Response: Thank you for your comments. We have modified this part of the content on 153-154 Lines. As follows: The relative extent of Z formation can be judged from the increase of the “darkbaseline” appar-ent after actinic off.
- I think that the authors made the incorrect measurements (as well as conclusions) ofΔpH because the normalization on flesh induced PSII+PSI signal is required (see New accessory for the DUAL-PAM-100: The P515/535 module and examples of its application, Ulrich Schreiber and Christof Klughammer, 2008) due to possible different content of PSII and/or I in leaves and bracts. In addition, how the authors can explain the lower ΔpH in leaves (L.163 and Fig 6A) but the significantly higher rate of CEF in leaves (L.102 and Fig. 3B), which should increase ΔpH.
Response: Thank you for your comments. The measurement method of proton gradient across thylakoid membrane was based on the literature of Ulrich Schreiber and Christof Klughammer, 2008, as well as the previous research of our laboratory (see Yi, X.P.; Zhang, Y.L.; Yao, H.S.; Han, J.M.; Chow, W.S.; Fan, D.Y.; Zhang, W.F. Changes in activities of both photosystems and the regulatory effect of cyclic electron flow in field-grown cotton (Gossypium hirsutum L) under water deficit. J. Plant Physiol. 2018, 220, 74–82. ), at the same time, it is measured by the program that comes with the system, which should not be wrong. And a recent paper reports that NPQ (Figure 6B) can indirectly reflect ΔpH (Figure 6A) (see Burlacot, A., O. Dao, P. Auroy, S. Cuine, Y. Li-Beisson & G. Peltier. (2022) Alternative photosynthesis pathways drive the algal CO(2)-concentrating mechanism. Nature 605, 366-371.), the trend of figure 6A and 6B in my experimental study is similar.
Previous studies have shown that cyclic electron flow can accelerate ΔpH, while my study has a higher cyclic electron flow and a weaker ΔpH in leaves. Respective contributions of cyclic electron flows around photosystem I for generating the proton gradient across the thylakoid membrane are ~30% (see Kawashima, R., R. Sato, K. Harada & S. Masuda. (2017) Relative contributions of PGR5- and NDH-dependent photosystem I cyclic electron flow in the generation of a proton gradient in Arabidopsis chloroplasts. Planta 246, 1045-1050.). So I think, first of all, cyclic electron flow is not the only condition to accelerate the generation of ΔpH, there are also linear electron transfer, and some alternative electron transfer pathways (see Burlacot, A., O. Dao, P. Auroy, S. Cuine, Y. Li-Beisson & G. Peltier. (2022) Alternative photosynthesis pathways drive the algal CO(2)-concentrating mechanism. Nature 605, 366-371.). Secondly, in my study, the ΔpH of leaves is lower than that of bracts. The higher ΔpH in the bracts may be generated by some other pathway, which will be further explored in the future. The above are just some of my views, I hope to get your approval. If we misunderstand the meaning of the reviewer, we can revise it again.
- I also doubt about NPQ results. The authors indicate that bracts had a considerably greater NPQ than leaves when light intensity increased, especially when there was a high light induction (L.168,169), however there are no statistically significant differences between NPQ induced in leaves and bracts at low and medium light (Fig. 6B left and center panels), while there is clear photoinhibition of PSII in bracts at high light (Fig. 6B right panel). Latter, of course, do not imply that the xanthophyll cycle's NPQ is driven by the ΔpH of the bracts (L170) as the authors write. See also in L205-206; L222-223; L232; L249.
Response: Thank you for your comments. According to the reviewer's opinion, we have modified relevant content on 167-175 Lines, as follows: According to further examination of the NPQ by fluorescence quenching curve, under conditions of low and medium light, the NPQ of bracts was marginally greater than that of leaves. However, NPQ of bracts was significantly higher than that of leaves at 2800 μmol m-2 s-1 high light (Figure 6B). The above showed that NPQ of bracts was sig-nificantly stronger under extremely high light conditions, accompanied by a strong ΔpH. These findings imply that NPQ is driven by the ΔpH of the bracts, that the xan-thophyll cycle may be involved. It appeared that PSI-CEF in cotton leaves and bracts could promote the establishment of ΔpH (Figure 6), and then drive NPQ to dissipate excess light energy, especially in bracts.
We have modified the content involved NPQ on 220-223 Lines, as follows: However, The generation of heat dissipation indicates that the photochemical activity of photosynthetic mechanism is weak and generates more excess excitation energy, on the other hand, it also indicates that the photosynthetic apparatus itself has the abil-ity to dissipate the excess excitation energy safely and harmlessly [36].
In my study, NPQ in bracts was significantly higher than in leaves during highlight induction, accompanied by high zeaxanthin synthesis and ΔpH. Since there is no data on Psbs protein, it is considered that the proton gradient may drive the heat dissipation of the lutein cycle. Based on the reviewer's comments, we have modified this content on results and discussion on 172-175 Lines, as follows: These findings imply that NPQ is driven by the ΔpH of the bracts, that the xanthophyll cycle may be involved. It appeared that PSI-CEF in cotton leaves and bracts could promote the establishment of ΔpH (Figure 6), and then drive NPQ to dissipate excess light energy, especially in bracts.
248-250 Lines, as follows: The ΔpH could establish to stimulate heat dissipation which maintained redox state of photosynthetic electron transport chain (Figure 7B).
- This is well known that the content of PTOX in thylakoids is very small, about 1% compared to PSII components (see DOI 10.1007/s00425-003-1111-7), thus the involvement of PTOX in described effects can not to be so sufficient (L215).
Response: Thank you for your comments. We have carefully read the literature you provided (DOI 10.1007/s00425-003-1111-7) and have also consulted relevant literature (DOI CNKI:SUN:ZWSL.0.2016-11-018, DOI 10.1016/j.molp.2016.06.008). PTOX chloroplast respiration is not the primary driver of transthylakoid membrane production. So we have modified this part of the content on 215-220 Lines. As follows: Studies had shown that alternative electron transport pathways, ion transporters/channels, pH-dependent and plastid terminal oxidase could also regulate the production of ΔpH, so whether the bracts have other electron transport pathways that is involved in the regulation of NPQ is worth further study.
- The authors should indicate the duration of saturation pulse for Fm detection (200-500 ms) (L385)
Response: Thank you for your comments. We have modified this part of the content on 385 Line. As follows: and a saturating light pulse light (20000 μmol m-1 s-1, the duration of saturated pulse Fm detection was 500 ms) was turned on every 30 s.
- L.259. () the author should insert the citations here.
Response: Thank you for your comments. We have modified this part of the content on 259 Line. As follows: According to earlier studies [5-8]
- L.357-358. Does Voltage mean value?
Response: Thank you for your comments. Figure 3B is the average of 3 biological repeats. The voltage of 0 s and 0.2 s in Figure. 3A are both a biological repeat.
- L.374-376. Confused sentences. Figure3A
Response: Thank you for your comments. We have rewritten this section on 376-380 lines. As follows: The dark-light-dark induced transients of 550-515 nm signal can reflect proton gradient across thylakoid membrane. Before the change of P515 signal was measured, the experimental material was dark adapted for a few hours, and then the actinic light (400 μmol m-2 s-1, 1800 μmol m-2 s-1 or 2800 μmol m-2 s-1) was turned on and induced for 10 minutes, and the actinic light was turned off and stopped when the P515 signal stabilized. (The first sentence explains that this curve can reflect proton gradient across thylakoid membrane, and the measurement process of the curve is introduced in the following sentence.)
Round 2
Reviewer 2 Report (New Reviewer)
I am grateful to the authors for their work with the manuscript in accordance with my previous comments and for their answers.
I still have some minor comments listed below. Nevertheless, I think that the manuscript can be accepted after minor revision.
From the response of the authors: ‘NdhM is a subunit of NDH complex, and the amount of its protein accumulation in Figure 2 can reflect the activity of NDH-mediated cyclic electron flow’. I think it will be useful to mention in the main text.
L94-95. ‘Lanes were loaded with thylakoid membrane proteins corresponding to 10 μg chlorophyll a’. → Lanes were loaded with thylakoid membrane proteins corresponding to 1 μg and 10 μg chlorophyll for Western analysis and CBB, respectively.
L155-156. ‘The signal above the black baseline gradually increased as the light intensity increased’. I still can not see the black baseline on the Fig. A. There are only green, blue, and red lines. The authors should add this black line on the Fig or change the description in the main text.
L350. chlorophyll a → chlorophyll
Author Response
Dear Editor,
We sincerely thank the reviewers for their comments regarding our manuscript. We have carefully revised the manuscript according to their suggestions. In our point-by-point response below, the reviewers’ comments are in Black and our responses are in BLUE. The modifications in the revised manuscript are in RED. We hope these modifications are acceptable to you. Thank you again for your time and consideration.
Reviewers/Editor comments:
1. From the response of the authors: ‘NdhM is a subunit of NDH complex, and the amount of its protein accumulation in Figure 2 can reflect the activity of NDH-mediated cyclic electron flow’. I think it will be useful to mention in the main text.
Response: Thank you for your comments. We have modified this part of the content on 84-86 Lines. As follows: Further, the expression of NdhM (NdhM is a subunit of NDH complex, and the amount of its protein accumulation can reflect the activity of NDH-mediated cyclic electron flow) and PGR5 proteins in leaves were higher (P < 0.05) than that in bracts (Figure 2B).
2. L94-95. ‘Lanes were loaded with thylakoid membrane proteins corresponding to 10 μg chlorophyll a’. → Lanes were loaded with thylakoid membrane proteins corresponding to 1 μg and 10 μg chlorophyll for Western analysis and CBB, respectively.
Response: Thank you for your comments. We have modified this part of the content on 94-95 Lines. As follows: Lanes were loaded with thylakoid membrane proteins corresponding to 1 μg and 10 μg chlorophyll for Western analysis and CBB, respectively.
3. L155-156. ‘The signal above the black baseline gradually increased as the light intensity increased’. I still can not see the black baseline on the Fig. 6A. There are only green, blue, and red lines. The authors should add this black line on the Fig or change the description in the main text.
Response: Thank you for your comments. “black baseline” is a proper noun. The increase in the light induced signal reflects the formation of zeaxanthin, the change in the light on to light off curve, which is called the “dark baseline” (27. Klughammer, C.; Schreiber, U. New accessory for the DUAL-PAM-100: The P515/535 module and examples of its application. PAM Appl. Notes 2008, 1, 1–10.).
We have modified this part of the content on 154-159 Lines. As follows: The formation of zeaxanthin can be reflected by an increase in light-induced signal. The formation of zeaxanthin is a slow process, formed gradually as protons accumulate in the thylakoid cavity. The relative extent of zeaxanthin formation can be judged from the increase of the “dark baseline” apparent after actinic off. Due to the high stability of the 550-515 nm signal difference, the slow change of the dark baseline can be used to reliably evaluate the reversible zeaxanthin content change.
4. L350. chlorophyll a → chlorophyll
Response: Thank you for your comments. We have modified this part of the content on 350 Lines. As follows: Lanes were loaded with thylakoid membrane proteins corresponding to 1 μg chlorophyll.
This manuscript is a resubmission of an earlier submission. The following is a list of the peer review reports and author responses from that submission.
Round 1
Reviewer 1 Report
In the paper entitled "Novel insights into the contribution of cyclic electron flow to 2 cotton bract in response to high light", the Authors describe an interesting and previously unrecognised role of cyclic electron flow around PSI (PSI-CEF) in bracts of Gossypium barbadense in response to high light. They compared PSI-CEF between leaves and bracts. I think the paper will be of interest to a wide audience of IJMS readers. Nevertheless, it needs improvement on a number of points. All the detailed comments can be found below.
The English style needs improvement. I think the text should be checked by a native English speaker.
Introduction
p. 2, l. 54: I think the better expression would be: “In contrast to the functional leaves, the non-leaved green organs usually have a lower photosynthetic capacity.”
p. 2, l. 60 – 63: The aim of the work is more or less defined, but a clearly formulated hypothesis is missing.
Results
p. 4, l. 71: What does ChlT stand for? Total chlorophyll (?). This is the first use and should be clarified.
p. 4, l. 94-95: I think it should be “cyclic electron transport rate”, not “cyclic electronic activity”
p.4, l. 96: add “and high light was…”
p. 6, l. 146: “Before the determination” of what?
p. 6, l. 150: It should be “the signal”
p. 7, l. 163-164: “significantly higher” (?). However, in Fig. 4B, statistical significance is not accurately indicated by the commonly used markers such as asterisks or letters. This should be indicated.
Discussion
p. 8, l. 181: “regulate” not “regulates”
p. 8, l. 182: “promotes” not “promote”
p. 8, l. 192: It was “enhanced”, but compared to what?
p. 8, l. 214: the phrase “the photochemical activity of photosynthetic mechanism” is unclear. Do you mean “photochemical activity of PSI”? This should be clarified.
p. 8, l. 225: “the excess excitation energy of the leaves”. This is unclear. Do you mean “the excess excitation energy absorbed by the leaves”? This should be clarified.
p. 8, l. 226: “photosynthetic mechanism” (?); I think it should be “photosynthetic apparatus”
p. 9, l. 249: “lead to the damage of active oxygen substances” (?). Not “active oxygen substances” but “active oxygen species” or better “reactive oxygen species (ROS)”. Furthermore, reactive oxygen species are not damaged, reactive oxygen species cause damage. This should be changed.
p. 9, l. 254-255: It should read “photosynthetic apparatus”
Materials and methods
p. 1. 287-288: The age of the plants on which the experiments were carried out and the average daylight intensity should be given.
p. 12, 4.5: Which primary and secondary antibodies were used (company ?) and in which dilution? This should be clarified.
p. 12, 4.6: Was the 'actinic light' a red light? Please state the wavelength of the "actinic light"
p. 13, 4.10: On the basis of how many repetitions were the mean values calculated (n = ?)?
Figures
Figure 2. The description of the figure should indicate what it represents: Mean and SD (SE) or maximum and minimum values and median or something else. It should also state on the basis of which number of repetitions (n =?) these graphs were made.
Figure 3. It should be stated how much protein was loaded onto the lanes.
Figure 4. What is meant by the small internal plot in Fig. 4A. This should be explained. The description should also state the number of repetitions (n =?) and that the bars represent the mean together with the SD (or SE).
Figure 5. If I understand correctly, the small area in Fig. 5A is shown on a larger scale in Fig. 5B. This should definitely be clarified. The time scale in Fig. 5B must be indicated. The number of repetitions (n =?) should also be stated in the description.
Figure 7. The description should also state the number of repetitions (n =?) and that the bars represent the mean together with the SD (or SE).
Figure 8. The arrow indicating water oxidation should touch PSII in a rounded rectangle. The thick blue line indicating CEF should have a marked arrowhead. The PGRL1 and NdhM proteins should also be shown in Fig 8.
Author Response
Reviewer #1: General comments
Specific comments
- line 54:I think the better expression would be: “In contrast to the functional leaves, the non-leaved green organs usually have a lower photosynthetic capacity.”
Response: Thank you for your comments. We have rewritten the sentence on 56-57 lines, as follow: The non-foliar green organs often have a lesser photosynthetic capacity than the functional leaves.
- line 60-63: The aim of the work is more or less defined, but a clearly formulated hypothesis is missing.
Response: Thank you very much for your comments. We have modified the scientific problem section on 61-64 lines. As follows: In response to excessive excitation energy, PSI-CEF can stimulate the heat dissipation process in cotton leaves. However, it is still unclear how non-photochemical quenching in cotton bracts drives heat dissipation that aids in the steady operation of photosynthetic machinery.
- line 71: What does ChlT stand for? Total chlorophyll (?). This is the first use and should be clarified.
Response: Thank you very much for your comments. We have modified this section on 71 line. As follows: ChlT (Total chlorophyll).
4 line 94-95: I think it should be “cyclic electron transport rate”, not “cyclic electronic activity”
Response: Thank you very much for your comments. We have modified this section on 98-101 lines. As follow: So the speed of the decline slope of the P700 reduction curve can indicates the rate of CEF, including NDH-mediated and PGR5-mediated PSI-CEF. On the whole, the CEF rate under medium light and high light was higher than that under low light, while the rate of CEF in leaves was significantly higher that in bracts.
5 line 96: add “and high light was…”
Response: Thank you very much for your comments. We have modified this section on lines 98-100. As follow: On the whole, the CEF rate under medium light and high light was higher than that under low light, while the rate of CEF in leaves was significantly higher that in bracts. Since the evidence of P700 dark reduction data in the highlight stage is not obvious, it is not specifically compared.
6 line 146: “Before the determination” of what?
Response: Thank you very much for your comments. We have modified that on 144 lines. As follow: This is a new module P515/535 of Dual-PAM-100 (Walz, Germany) was used to detect the differential absorption of 550-515 nm measuring light. Before actinic light (Before AL): cotton leaves and bracts need to be dark acclimated for 1 hour to reflect the integrity of the thylakoid membrane. After actinic light (After AL): the actinic light was turned on for 10 min, and then turned off, followed by dark adaptation for 4 min before measurement. Before AL: measurement of leaves and bracts after 1 h dark adaptation.
7 line 150: It should be “the signal”
Response: Thank you very much for your comments. We have modified that on line 154. As follow: The slow increase above the signal line can reflect the formation and change of zeaxanthin. Transmembrane potential (Δψ) and pH can be determined.
8 line 163-164: “significantly higher” (?). However, in Fig. 4B, statistical significance is not accurately indicated by the commonly used markers such as asterisks or letters. This should be indicated.
Response: Thank you very much for your comments. Difference analysis with a significance level of P < 0.05 were carried out and marked with lowercase English letters in the figure. The data in Fig. 4B is indirect and mainly describe the proton gradient across the thylakoid membrane (ΔpH) in Fig 7A, showing obvious higher, we have modified that on 158-161 lines. As follows: It can be seen from Fig. 6A that with the increase of light intensity from low light to high light, the ΔpH gradually increased, especially in the middle and high light intensity, and that the ΔpH of the leaves was significantly lower than that of the bract leaves.
9 line 181: “regulate” not “regulates”
Response: Thank you very much for your comments. “regulates” has been modified into “regulate” on 181 line. As follows: CEF can regulate the redox state of the photosynthetic electron transport chain.
10 line 182: “promotes” not “promote”
Response: Thank you very much for your comments. “promote” has been modified into “promotes” on 182 lines. As follows: promotes the establishment of a ΔpH across the thylakoid membrane.
11 line 192: It was “enhanced”, but compared to what?
Response: Thank you very much for your comments. We have modified this section on 198-200 lines. As follows: The rate of cyclic electron flow was increased in cotton leaves as the light intensity steadily increased from low light to high light, and the activity of ATP synthase was also increased.
12 line 214: the phrase “the photochemical activity of photosynthetic mechanism” is unclear. Do you mean “photochemical activity of PSI”? This should be clarified.
Response: Thank you very much for your comments. Yes, it should be the photochemical activity of PSII. We have modified that on 215-217 lines. As follows: “the photochemical activity of photosynthetic mechanism” means photochemical activity of PSII which is the ability to absorb and convert light energy.
13 line 225: “the excess excitation energy of the leaves”. This is unclear. Do you mean “the excess excitation energy absorbed by the leaves”? This should be clarified.
Response: Thank you very much for your comments. Yes, it is “the excess excitation energy absorbed by the leaves”. We have modified this section on 227-228 lines. As follows: It can be speculated that under high light condition, although the excess excitation energy aborbed by the leaves increased.
14 line 226: “photosynthetic mechanism” (?); I think it should be “photosynthetic apparatus”
Response: Thank you very much for your comments. Yes, it is “photosynthetic apparatus”. We have modified this section on 229 line. As follows: the ability of the leaves to protect the photosynthetic apparatus through heat dissipation in high light is weakened.
15 line 249: “lead to the damage of active oxygen substances” (?). Not “active oxygen substances” but “active oxygen species” or better “reactive oxygen species (ROS)”. Furthermore, reactive oxygen species are not damaged, reactive oxygen species cause damage. This should be changed.
Response: Thank you very much for your comments. We have modified this section on 258-259 lines. As follows: the light captured by the leaves is not all used for carbon assimilation, excess light energy lead to the production of active oxygen substances (ROS), which harms the photosynthetic organs and affect the stable operation of leaf photosynthesis, and even affect the yield of cotton.
16 line 254-255: It should read “photosynthetic apparatus”
Response: Thank you very much for your comments. Yes, it is. We have modified this section on 263 lines. A 5s follows: Bracts of non-foliar green organisms have a fully functional photosynthetic apparatus and function, as well as a strong capacity to withstand adversity stress, according to previous research and our findings.
17 line 287-288: The age of the plants on which the experiments were carried out and the average daylight intensity should be given.
Response: Thank you very much for your comments. The experimental materials were carried out under the light intensity of 1500-2000 μmol m-2 s-1 at the maturity stage of cotton. We have added the information on 297-299 lines, as follows: The experimental materials were carried out under the light intensity of 1500-2000 μmol m-2 s-1 at the maturity stage of cotton. The first fully expanded leaves and the bracts corresponding to the first fruit node of each cotton plant were sampled three times.
18 4.5: Which primary and secondary antibodies were used (company ?) and in which dilution? This should be clarified.
Response: Thank you very much for your comments. Primary and secondary antibodies were provided by line 320-323. As follows:
Protein western blotting was performed after electrophoresis. The proteins in the polyacrylamide gel were electrotransferred to to a polyvinylidene difluoride membrane (Immobilon-P; Millipore) and the PVDF membrane was placed in 5% skim milk prepared by TBST and closed in a shak for 2 hours at room temperature. The corresponding protein-specific antibodies were incubated in the shak for 2 hours, and then TBST was used to wash the antibodies twice for 12 minutes each time. Then, the corresponding dilution of the second antibody was added and incubated for 1 hour. TBST was used to wash the antibody for 4 times, each time for 12 minutes. The development test was performed according to the ECL assay kit (Amersham Pharmacia). Antibodies against NAD(P)H-quinone oxidoreductase subunit M (NdhM) and PGR5-LIKE PHOTOSYNTHETIC PHENOTYPE1 (PGRL1)were provided by Professor Lianwei Peng (College of Life Sciences, Shanghai Normal University).
This experiment was carried out with the help of Ma Weimin Laboratory, College of Life Sciences, Shanghai Normal University.
19 4.6: Was the 'actinic light' a red light? Please state the wavelength of the "actinic light"
Response: Thank you very much for your comments. We have modified that on 359 lines. As follows: Leaves and bracts were exposed to actinic light (620nm, 400 μmol m-2 s-1, 1800 μmol m-2 s-1 or 2800 μmol m-2 s-1).
20 4.10: On the basis of how many repetitions were the mean values calculated (n = ?)?
Response: Thank you very much for your comments. There were three repetitions to be calculated the mean values. We have added some information in “Statistical analysis” section and the notes of figures and tables.
21 Figure 2. The description of the figure should indicate what it represents: Mean and SD (SE) or maximum and minimum values and median or something else. It should also state on the basis of which number of repetitions (n =?) these graphs were made.
Response: Thank you very much for your comments. We have modified that in the title of figure 2. As follows: Solid within the box indicate the mean values, box boundaries indicate upper and lower quartiles; The width of spindle denote the accumulation of high samples. Bars indicate SD (n = 3).
22 Figure 3. It should be stated how much protein was loaded onto the lanes.
Response: Thank you very much for your comments. We have modified that in the title of figure 3. As follows: Antibodies against NAD(P)H-quinone oxidoreductase subunit M (NdhM) (1000:1) and PGR5-LIKE PHOTOSYNTHETIC PHENOTYPE1 (PGRL1) (1000:1) were provided by Professor Lianwei Peng (College of Life Sciences, Shanghai Normal University).
23 Figure 4. What is meant by the small internal plot in Fig. 4A. This should be explained. The description should also state the number of repetitions (n =?) and that the bars represent the mean together with the SD (or SE).
Response: Thank you very much for your comments. We have modified the figure 4 and added some information in the title to explain the figure on 102-103 lines. As follows: Different letters indicate a significant difference (p < 0.05) between the low, medium and high light. ** denotes significant at the 0.01 level between the leaves and bracts based on analysis of variance. Bars indicate SD (n = 3).
24 Figure 5. If I understand correctly, the small area in Fig. 5A is shown on a larger scale in Fig. 5B. This should definitely be clarified. The time scale in Fig. 5B must be indicated. The number of repetitions (n =?) should also be stated in the description.
Response: Thank you very much for your comments. We have modified that in Figure 5 and the title of the figure on 102-103 lines. As follows: Monitoring of NDH activity (red circle below Fig. 5B) by using chlorophyll fluorescence analysis in cotton leaves and bracts (n=3) under different light intensities of 400 μmol m-2 s-1 (low light), 1800 μmol m-2 s-1 (medium light) or 2800 μmol m-2 s-1 (high light). (A) Induction kinetics and post-illumination transient in chlorophyll fluorescence, AL (actinic light 400 μmol m-2 s-1, 1800 μmol m-2 s-1 or 2800 μmol m-2 s-1, lasted for 4min), ML (measuring light). (B) Analyses of post-illumination chlorophyll fluorescence transient in cotton leaves and bracts.
25 Figure 7. The description should also state the number of repetitions (n =?) and that the bars represent the mean together with the SD (or SE).
Response: Thank you very much for your comments. We have added the information in the title of Figure 7 about “Bars indicate mean and SD (n = 3).”
26 Figure 8. The arrow indicating water oxidation should touch PSII in a rounded rectangle. The thick blue line indicating CEF should have a marked arrowhead. The PGRL1 and NdhM proteins should also be shown in Fig 8.
Response: Thank you very much for your comments. We have modified the figure according to your comments. As follows:

Reviewer 2 Report
The present manuscript compares various aspects of cyclic electron transfer in the bracts of cotton to those of leaves. The topic is interesting, as we still do not have much knowledge about the detailed photosynthetic properties of non-foliar green tissues. However, the quality of scientific English used throughout the manuscript makes reading difficult and reasoning behind the conclusions remains unclear. Thus, the language and the structure of the paper should be significantly improved.
Introduction: Describe the distinct pathways of cyclic electron transfer, introduce non-photochemical quenching and comparison of leaf-bract biology.
Materials and methods: More details are needed. For instance, how is the sampling conducted? How many leaves (in one or several plants?) in one biological replicate? At what time of day were the samples taken? Although coordinates of the research station are provided, it would be informative to know something about the environmental conditions, including light. Practically nothing is told about the RNA-Seq analysis, much more information is needed. More information also needed for chloroplast isolation, Western blotting (e.g. info about the antibodies) etc etc
Results: If RNA-Seq analysis was performed, all data should be shown. Were the differences shown in Fig. 2 significant? Detailed statistical analysis is lacking. For the Western blot, more information should be provided. Were the gel loaded on Chl or protein bases? How much was loaded? What kind of gel was used? Is Fig 4A presenting results from leaves or bracts? etc etc
Discussion: The conclusions are not convincingly supported by the results.
Author Response
Dear Editor,
We sincerely thank the reviewers for their comments regarding our manuscript. We have carefully revised the manuscript according to their suggestions. In our point-by-point response below, the reviewers’ comments are in Black and our responses are in BLUE. The modifications in the revised manuscript are in RED. We hope these modifications are acceptable to you. Thank you again for your time and consideration.
Reviewer #2:
Some specific comments are given below:
1 The present manuscript compares various aspects of cyclic electron transfer in the bracts of cotton to those of leaves. The topic is interesting, as we still do not have much knowledge about the detailed photosynthetic properties of non-foliar green tissues. However, the quality of scientific English used throughout the manuscript makes reading difficult and reasoning behind the conclusions remains unclear. Thus, the language and the structure of the paper should be significantly improved.
Response: Thank you very much for your comments. We have revised and polished the language. The language of the paper have been edited for native English speaking editors at AJE (https://www.aje.cn/). The paper have been carefully modified according to the comments of Editor and Reviewers, we hope these modifications are acceptable to you. Thank you again for your time and consideration.
2 Introduction: Describe the distinct pathways of cyclic electron transfer, introduce non-photochemical quenching and comparison of leaf-bract biology.
Response: Thank you very much for your comments. We have introduce some in formation about “the distinct pathways of cyclic electron transfer” and “introduce non-photochemical quenching and comparison of leaf-bract biology” on 215-316 lines in Introduction section. As follows:
Cyclic electron flow around PSI (PSI-CEF) as a photoprotective mechanism that has received much attention in recent years, which mainly through PROTON GRADIENT REGULATION5 (PGR5) and choroplastic NAD(P)H dehydrogenase (NDH) complextwo pathways. Studies have demonstrated that the NDH and PGR5 pathways play a critical part in the process of resisting adverse environmental conditions because they form two complementary cyclic electron transport pathways that cannot both be absent at the same time. So far fewer investigations have been conducted on the cyclic electron flow-related characteristics in crop non-foliar green organs than have been done on the regulatory role of cyclic electron transfer in cotton leaves.
The non-foliar green organs often have a lesser photosynthetic capacity than the functional leaves. Non-foliar green organs will therefore experience higher excess excitation energy at the same light intensity. In addition, Wullschleger et al. shown that under conditions of environmental stress, the relative contribution rate of cotton bracts to yield will rise. This further demonstrated the significance of non-foliar green organs' photoprotective mechanisms. In response to excessive excitation energy, PSI-CEF can stimulate the heat dissipation process in cotton leaves. However, it is still unclear how non-photochemical quenching in cotton bracts drives heat dissipation that aids in the steady operation of photosynthetic machinery.
3 Materials and methods: More details are needed. For instance, how is the sampling conducted? How many leaves (in one or several plants?) in one biological replicate? At what time of day were the samples taken? Although coordinates of the research station are provided, it would be informative to know something about the environmental conditions, including light. Practically nothing is told about the RNA-Seq analysis, much more information is needed. More information also needed for chloroplast isolation, Western blotting (e.g. info about the antibodies) etc etc
Response: Thank you very much for your comments. We have modified that in the section, as follows:
- Since most of the experiments involve the measurement of chlorophyll fluorescence, it is best to select the leaves and bracts before dawn for the experimental materials, because the light intensity at this time is weak and will not cause light influence on the experimental materials.
- Line 291-292, The experimental materials were carried out under the light intensity of 1500-2000 μmol m-2s-1 at the maturity stage of cotton. The first fully expanded leaves and the bracts corresponding to the first fruit node of each cotton plant were sampled three times.
- RNA-Seq analysis, as follows:
Cotton leaves and bracts were taken under growth light conditions. Each group contained 3 replicates, totaling 6 samples. The selection of experimental materials shown in the Fig. 1. The processing of cotton leaves and bracts, including RNA extraction, RNA quality detection, RNA library construction and sequencing, were completed by Beijing Novogene Bioinformation Technology.
- Illumina HiSep 4000 was used to double-terminally sequence the RNA samples. The height of transcripts in the sample is indicated by the FPKM (Fragments Per Kilobase of Transcript per Million Fragments Mapped) number. Set the reference standard expression differences (FC, the fold change) were greater than 4, or | FC|> 4 and p.adjust<0.05 was the significantly differentially expressed gene.
- Western blotting as follows:
4.5. Sodium dodecyl sulfate polyacrylamide gel electrophoresis (SDS-PAGE)
Chloroplasts were osmotically ruptured in buffer containing 20 mM HEPES/KOH (pH 7.6), 5 mM MgCl2 and 2.5 mM EDTA. Thylakoid membranes were separated from the stromal fraction by centrifugation (15,000 ×g for 10 min at 4°C). The chlorophyll content was determined by the method of Porra et al. [23].
SDS-PAGE of thylakoid membranes was carried out on 12% polyacrylamide gel with 6 M urea as described earlier [24]. The coarse thylakoids extracted in accordance with 4.4 were resuspended with Medium A buffer solution, adjusted to the same chlorophyll concentration, and then divided into 1.5mL centrifugal-tubes according to the required loading amount. 1/4 volume of 5x SDS-loading buffer (2 M urea, 0.5 M Tris-HCl, respectively) was added. pH 8.0, 20% glycerin, 7.5% SDS, 2% (v/v) mercaptoethanol, 0.05% (w/v) bromophenol blue) were mixed and boiled in a water bath at 100℃ for 8 min, and then centrifuged at room temperature at 12000 rpm for 10 min. Supernatant was taken into a new centrifugal tube and samples were taken. Wait for the sample to run out of the glue at a voltage of 60 V and adjust the voltage to 80 V until the end of electrophoresis.
4.6. Western blotting analysis of proteins
Protein western blotting was performed after electrophoresis. The proteins in the polyacrylamide gel were electrotransferred to to a polyvinylidene difluoride membrane (Immobilon-P; Millipore) and the PVDF membrane was placed in 5% skim milk prepared by TBST and closed in a shak for 2 hours at room temperature. The corresponding protein-specific antibodies were incubated in the shak for 2 hours, and then TBST was used to wash the antibodies twice for 12 minutes each time. Then, the corresponding dilution of the second antibody was added and incubated for 1 hour. TBST was used to wash the antibody for 4 times, each time for 12 minutes. The development test was performed according to the ECL assay kit (Amersham Pharmacia). Antibodies against NAD(P)H-quinone oxidoreductase subunit M (NdhM) (1:1000) and PGR5-LIKE PHOTOSYNTHETIC PHENOTYPE1 (PGRL1) (1:1000) were provided by Professor Lianwei Peng (College of Life Sciences, Shanghai Normal University).
This experiment was carried out with the help of Ma Weimin Laboratory, College of Life Sciences, Shanghai Normal University.
4 Results: If RNA-Seq analysis was performed, all data should be shown. Were the differences shown in Fig. 2 significant? Detailed statistical analysis is lacking. For the Western blot, more information should be provided. Were the gel loaded on Chl or protein bases? How much was loaded? What kind of gel was used? Is Fig 4A presenting results from leaves or bracts? etc etc
Response: Thank you very much for your comments. We have modified modified the figure 2 and the statistical analysis also was added in the figure. The information about Western blot were also provided in Materials and methods section and the title of the figure 2. As follows: on 86-93 lines in Materials and methods section.
5 Discussion: The conclusions are not convincingly supported by the results.
Response: Thank you very much for your comments. We have modified that in Discussion section on 125-126 lines. As follows: The protective mechanism of cotton leaves can be mainly reflected in the curve of ATP synthesis in Fig. 6, while the data of bracts can be mainly reflected in Fig 7A,B. We are sorry that due to the limited experimental conditions, we can only detect physiological data. By summarizing the previous studies, the relevant data of chlorophyll fluorescence have a certain reliability (Fig 6, Fig 7).

Reviewer 3 Report
A review of the article entitled “Novel insights into the contribution of cyclic electron flow to cotton bract in response to high light”.
The authors compered parameters related to cyclic electron flow in cotton bracts and leaves at different light intensities ( low, medium and high). The NDH activity and non-potochemical quenching was measured using a chlorophyll fluorescence. Also ATP activity measurements were performed based of changes on P515 signal. The expression level of PGR5, PGRL and NADH genes was investigated as well as the protein level of NdhM and PGR5 and pigment content was analysed.
In bracts,in all investigated light intensities authors observed increased, in relation to leaves, activity of NDH -dependent pathway of cyclic electron flow. Simultaneously the activity of ATPase in bracts deacrease with increase of light intensity. What is more obtained results indicated that more zeaxanthin is synthetised in bracts than in leaves. Authors conclude that in bracts photoprotection mechanisms are activated leading to high NPQ and low ATP. Author suggest also that bracts are characterised by strong resistance to stress.
The manuscript contains interesting results and experiments were well planed and performed correctly, however some issues need to be improved or however clarified.
The “Introduction” is short and insufficient. The information about PS-CEF1 pathway, its mechanism and role should be supplemented (for example the presence of NDH and PGR5 dependent mechanism is mention only in the “Results” section). Authors should also include more detailed information concerning present knowledge about photosynthesis in cotton bracts. Also in the “Discussion” section the results should be more clearly discussed in context of previous findings on photosynthetic machinery and reactions in cotton bracts.
The statement “Therefore, bracts may mainly rely on the establishment of ΔpH to stimulate heat dissipation via xanthophylls cycle (p.8 line 202) needs clarification.
The fragment “ Under the same light intensity, bracts have more excess excitation energy and stronger heat dissipation capacity to maintain the stable operation of photosynthetic apparatus. It can be seen from Fig. 4, 5, and 7 that under both low light and high light, ΔpH, heat dissipation and the cyclic electron activity all increased, indicating that the bracts mainly rely on the ΔpH established by cyclic electrons to drive the NPQ to dissipate excess light energy.” (p.8 lines 218-222) needs clarification
In „Material and methods” section the section 4.3 Transcriptome sequencing should be described in greater detail.
I suggest also to split 4.5 section. Thylakoid membrane protein preparation and immunoblot analysis) on two separate sections. How were the samples prepared for the electrophoresis, how much sample was applied on gel? what method of antibodies detection was applied?
In section 4.8 a reference is missing.
Please rewrite the “Conclusion” section since it is hard to understand.
I strongly recommend language proofreading of the manuscript.
In conclusion. I recommend accept the manuscript after the minor corrections.
Author Response
Dear Editor,
We sincerely thank the reviewers for their comments regarding our manuscript. We have carefully revised the manuscript according to their suggestions. In our point-by-point response below, the reviewers’ comments are in Black and our responses are in BLUE. The modifications in the revised manuscript are in RED. We hope these modifications are acceptable to you. Thank you again for your time and consideration.
Reviewer #3:
A review of the article entitled “Novel insights into the contribution of cyclic electron flow to cotton bract in response to high light”.
The authors compered parameters related to cyclic electron flow in cotton bracts and leaves at different light intensities ( low, medium and high). The NDH activity and non-potochemical quenching was measured using a chlorophyll fluorescence. Also ATP activity measurements were performed based of changes on P515 signal. The expression level of PGR5, PGRL and NADH genes was investigated as well as the protein level of NdhM and PGR5 and pigment content was analysed.
In bracts,in all investigated light intensities authors observed increased, in relation to leaves, activity of NDH -dependent pathway of cyclic electron flow. Simultaneously the activity of ATPase in bracts deacrease with increase of light intensity. What is more obtained results indicated that more zeaxanthin is synthetised in bracts than in leaves. Authors conclude that in bracts photoprotection mechanisms are activated leading to high NPQ and low ATP. Author suggest also that bracts are characterised by strong resistance to stress.
The manuscript contains interesting results and experiments were well planed and performed correctly, however some issues need to be improved or however clarified.
1 The “Introduction” is short and insufficient. The information about PS-CEF1 pathway, its mechanism and role should be supplemented (for example the presence of NDH and PGR5 dependent mechanism is mention only in the “Results” section). Authors should also include more detailed information concerning present knowledge about photosynthesis in cotton bracts. Also in the “Discussion” section the results should be more clearly discussed in context of previous findings on photosynthetic machinery and reactions in cotton bracts.
Response: Thank you very much for your comments. We have added some information about research tends of cyclic electron transport pathways in the Introduction section on 125-126 lines. As follows:
Cyclic electron flow around PSI (PSI-CEF) as a photoprotective mechanism that has received much attention in recent years [1–3], which mainly through PROTON GRADIENT REGULATION5 (PGR5) [4, 33] and choroplastic NAD(P)H dehydrogenase (NDH) complex [25, 34, 35] two pathways. Studies have demonstrated that the NDH and PGR5 pathways play a critical part in the process of resisting adverse environmental conditions because they form two complementary cyclic electron transport pathways that cannot both be absent at the same time.
So far fewer investigations have been conducted on the cyclic electron flow-related characteristics in crop non-foliar green organs than have been done on the regulatory role of cyclic electron transfer in cotton leaves.
2 The statement “Therefore, bracts may mainly rely on the establishment of ΔpH to stimulate heat dissipation via xanthophylls cycle (p.8 line 202) needs clarification.
Response: Thank you very much for your comments. We have modified this section on lines 208-210. As follows: Correspondingly, under medium and high light conditions the ΔpH and zeaxanthin synthesis rates of bracts were significantly higher than those of leaves (Fig. 6). Therefore, bracts may mainly rely on the establishment of ΔpH to stimulate heat dissipation.
3 The fragment “ Under the same light intensity, bracts have more excess excitation energy and 3 stronger heat dissipation capacity to maintain the stable operation of photosynthetic apparatus. It can be seen from Fig. 4, 5, and 7 that under both low light and high light, ΔpH, heat dissipation and the cyclic electron activity all increased, indicating that the bracts mainly rely on the ΔpH established by cyclic electrons to drive the NPQ to dissipate excess light energy.” (p.8 lines 218-222) needs clarification
Response: Thank you very much for your comments. We have modified the sentence on lines. As follows: Due to the low photosynthetic capacity of bracts, they will experience greater exciation energy at the same level of light intensity, so bracts stimulate stronger heat dissipation capacity to maintain the stable operation of photosynthetic apparatus. It can be seen from Fig. 4, 5, and 7 show that ΔpH, heat dissipation, and cyclic electron activity all increased in response to low light and high light, respectively. This suggests that the bracts primarily depend on the ΔpH established by cyclic electrons to drive the NPQ to dissipate excess light energy.
4 In „Material and methods” section the section 4.3 Transcriptome sequencing should be described in greater detail.
Response: Thank you very much for your comments. We have modified this section on 310-320 lines. As follows:
4.3. Transcriptome sequencing
Cotton leaves and bracts were taken under growth light conditions. Each group contained 3 replicates, totaling 6 samples. The selection of experimental materials shown in the Fig. 1. The processing of cotton leaves and bracts, including RNA extraction, RNA quality detection, RNA library construction and sequencing, were completed by Beijing Novogene Bioinformation Technology.
Illumina HiSep 4000 was used to double-terminally sequence the RNA samples. The height of transcripts in the sample is indicated by the FPKM (Fragments Per Kilobase of Transcript per Million Fragments Mapped) number. Set the reference standard expression differences (FC, the fold change) were greater than 4, or | FC|> 4 and p.adjust<0.05 was the significantly differentially expressed gene.
5 I suggest also to split 4.5 section. Thylakoid membrane protein preparation and immunoblot analysis) on two separate sections. How were the samples prepared for the electrophoresis, how much sample was applied on gel? what method of antibodies detection was applied?
In section 4.8 a reference is missing.
Response: Thank you very much for your comments. We have modified this section on 328-357 lines. As follows:
4.5. Sodium dodecyl sulfate polyacrylamide gel electrophoresis (SDS-PAGE)
Chloroplasts were osmotically ruptured in buffer containing 20 mM HEPES/KOH (pH 7.6), 5 mM MgCl2 and 2.5 mM EDTA. Thylakoid membranes were separated from the stromal fraction by centrifugation (15,000 ×g for 10 min at 4°C). The chlorophyll content was determined by the method of Porra et al. [23].
SDS-PAGE of thylakoid membranes was carried out on 12% polyacrylamide gel with 6 M urea as described earlier [24]. The coarse thylakoids extracted in accordance with 4.4 were resuspended with Medium A buffer solution, adjusted to the same chlorophyll concentration, and then divided into 1.5mL centrifugal-tubes according to the required loading amount. 1/4 volume of 5x SDS-loading buffer (2 M urea, 0.5 M Tris-HCl, respectively) was added. pH 8.0, 20% glycerin, 7.5% SDS, 2% (v/v) mercaptoethanol, 0.05% (w/v) bromophenol blue) were mixed and boiled in a water bath at 100℃ for 8 min, and then centrifuged at room temperature at 12000 rpm for 10 min. Supernatant was taken into a new centrifugal tube and samples were taken. Wait for the sample to run out of the glue at a voltage of 60 V and adjust the voltage to 80 V until the end of electrophoresis.
4.6. Western blotting analysis of proteins
Protein western blotting was performed after electrophoresis. The proteins in the polyacrylamide gel were electrotransferred to to a polyvinylidene difluoride membrane (Immobilon-P; Millipore) and the PVDF membrane was placed in 5% skim milk prepared by TBST and closed in a shak for 2 hours at room temperature. The corresponding protein-specific antibodies were incubated in the shak for 2 hours, and then TBST was used to wash the antibodies twice for 12 minutes each time. Then, the corresponding dilution of the second antibody was added and incubated for 1 hour. TBST was used to wash the antibody for 4 times, each time for 12 minutes. The development test was performed according to the ECL assay kit (Amersham Pharmacia). Antibodies against NAD(P)H-quinone oxidoreductase subunit M (NdhM) (1000:1) and PGR5-LIKE PHOTOSYNTHETIC PHENOTYPE1 (PGRL1) (1000:1) were provided by Professor Lianwei Peng (College of Life Sciences, Shanghai Normal University).
This experiment was carried out with the help of Ma Weimin Laboratory, College of Life Sciences, Shanghai Normal University.
6 Please rewrite the “Conclusion” section since it is hard to understand.
Response: Thank you very much for your comments. We have rewriten the Conclusion on 311-320 lines. As follows: Cotton leaves and bracts play different photoprotective mechanisms in response to high light. Leaves mainly rely on cyclic electron flow to drive ATP synthase to synthesize ATP and optimize ATP/NADPH for efficient photosynthesis. On contrast, bracts mainly can protect the photosynthesis by establishing a ΔpH through cyclic electron flow to stimulate the process of heat dissipation, Especially in the high light, the heat dissipation ability of bracts can be continuously enhanced, while this was not observed in the leaves. While this was not apparent in the leaves, bracts can continuously increase their capacity to dissipate heat, especially in intense light. So bracts have comparatively steady photosynthetic capability and stronger stress tolerance.

Round 2
Reviewer 1 Report
The authors have made most of the suggested changes. Nevertheless, I would like to make a few comments on the revised manuscript below:
1. There is no indication from the authors that the English was checked and corrected by a native speaker. But OK, the linguistic side is acceptable.
2. The authors have not formulated a clear hypothesis, but the aim of the paper is acceptable.
3. Line 71: "total chlorophyll", not "Total chlorophyll”.
4. Figure 3: It is not a "black circle", but a "black rectangle".
5. Figure 4: It is not a "red circle" but a "red rectangle".
6. Line 258: "reactive oxygen species (ROS)", not "reactive oxygen substances".
7. Lines: 298 - 299: The light intensity indicated is probably the high light to which the plants were exposed. The light in which the plants grew is not given. Did the authors measure the intensity of this light at all?
8. Line 341: The centrifugation speed should be given in "g" and not in “rpm”
9. Lines: 352 -356: It is only stated which primary antibodies were used. The dilution of the anti-NdhM and anti-PGRL1 antibodies is given as 1000 : 1, whereas in the description of Fig. 2B it is 1:1000. Which dilution is correct? This should be clarified.
10. In my opinion, the description of Fig. 2B refers to the dilution of antibodies and not to the amount of total protein in the sample loaded into the gel lane. This should be clarified.
Author Response
Dear Editor,
We sincerely thank the reviewers for their comments regarding our manuscript. We have carefully revised the manuscript according to their suggestions. In our point-by-point response below, the reviewers’ comments are in Black and our responses are in BLUE. The modifications in the revised manuscript are in RED. We hope these modifications are acceptable to you. Thank you again for your time and consideration.
Reviewers/Editor comments:
Reviewer #1: General comments
Specific comments
The authors have made most of the suggested changes. Nevertheless, I would like to make a few comments on the revised manuscript below:
1. There is no indication from the authors that the English was checked and corrected by a native speaker. But OK, the linguistic side is acceptable.
Response: Thank you very much for your comments. The language of the paper have been edited for native English speaking editors at AJE (https://www.aje.cn/). The paper have been carefully modified according to the comments of Editor and Reviewers, we hope these modifications are acceptable to you. Thank you again for your time and consideration.
- The authors have not formulated a clear hypothesis, but the aim of the paper is acceptable.
Response: Thank you very much for your comments. We have modified this section on 62-65 lines. As follows:
However, the mechanism of non-photochemical quenching in cotton bracts drives heat dissipation
assist the steady operation of photosynthetic machinery is still unclear. Thus, it also explores the underlying mechanisms of photoprotection in cotton bracts by comparative analyzing the changes of the CEF-PSI, proton gradient across the thylakoid membrane and heat dissipation in cotton leaf and bract under different light intensity.
- Line 71: "total chlorophyll", not "Total chlorophyll”.
Response: Thank you very much for your comments. We have modified "Total chlorophyll” into "total chlorophyll" on 71 line.
- Figure 3: It is not a "black circle", but a "black rectangle".
Response: Thank you very much for your comments. We have modified that on 104 line. As follows: black rectangle reflects the initial reduction rate (0-0.2 s) of P700+ after turning off the far red light (FR) as follow Fig 3B.
- Figure 4: It is not a "red circle" but a "red rectangle".
Response: Thank you very much for your comments. We have modified that on 119 line. As follows: Monitoring of NDH activity (red rectangle below Fig. 4B) by using chlorophyll fluorescence analysis in cotton leaves and bract
6. Line 258: "reactive oxygen species (ROS)", not "reactive oxygen substances".
Response: Thank you very much for your comments. We have modified "reactive oxygen substances" into "reactive oxygen species (ROS)" on 257 line. As follows:
.....excess light energy lead to the production of active oxygen species (ROS), which harms the photosynthetic organs.
- Lines: 298 - 299: The light intensity indicated is probably the high light to which the plants were exposed. The light in which the plants grew is not given. Did the authors measure the intensity of this light at all?
Response: Thank you very much for your comments. We have modified that on 298-300 lines. As follows: The experimental materials were sampled at the maturity stage, the highest light intensity is in the range of 1500 to 2000 mol m-2 s-1 at the stage.
8. Line 341: The centrifugation speed should be given in "g" and not in “rpm”
Response: Thank you very much for your comments. We have modified that on 341 line. As follows:
then centrifuged at room temperature at 12000 g for 10 min.
9. Lines: 352 -356: It is only stated which primary antibodies were used. The dilution of the anti-NdhM and anti-PGRL1 antibodies is given as 1000 : 1, whereas in the description of Fig. 2B it is 1:1000. Which dilution is correct? This should be clarified.
Response: Thank you very much for your comments. We have modified this section on 354-355 lines. As follows: Antibodies against NAD(P)H-quinone oxidoreductase subunit M (NdhM) (1:1000) and PGR5-LIKE PHOTOSYNTHETIC PHENOTYPE1 (PGRL1) (1:1000) were provided by Professor Lianwei Peng (College of Life Sciences, Shanghai Normal University).
- In my opinion, the description of Fig. 2B refers to the dilution of antibodies and not to the amount of total protein in the sample loaded into the gel lane. This should be clarified.
Response: Thank you very much for your comments. We have modified this section on 93-94 lines. As follows: Coomassie Brilliant Blue (CBB)-staining PAGE profiles and Western analysis of total PGR5-LIKE PHOTOSYNTHETIC PHENOTYPE1 (PGRL1) (1:1000) and NAD(P)H-quinone oxidoreductase subunit M (NdhM) (1:1000) in the thylakoid membranes of cotton leaves and bracts. Lanes were loaded with thylakoid membrane proteins corresponding to 1 μg chlorophyll a.

Round 3
Reviewer 1 Report
The authors have incorporated the amendments.
A small comment:
l. 264, the same comment as before: "reactive oxygen species (ROS)" instead of "active oxygen substances" (!)
Author Response
Dear Editor,
We sincerely thank the reviewers for their comments regarding our manuscript. We have carefully revised the manuscript according to their suggestions. In our point-by-point response below, the reviewers’ comments are in Black and our responses are in BLUE. The modifications in the revised manuscript are in RED. We hope these modifications are acceptable to you. Thank you again for your time and consideration.
Reviewers/Editor comments:
Reviewer #1: General comments
264, the same comment as before: "reactive oxygen species (ROS)" instead of "active oxygen substances" (!)
Response: Thank you very much for your comments. We have modified that on 264 line.
